# An atomically controlled insulator-to-metal transition in iridate/manganite heterostructures

Enyang Men[1,2], Deyang Li[1,2], Haiyang Zhang[1,2], Jingxin Chen[1,2], Zhihan Qiao[1,2], Long Wei ®[3], Zhaosheng Wang[1], Chuanying Xi[1], Dongsheng Song ®[4], Yuhan Li[5], Hyoungjeen Jeen ®[6], Kai Chen ®[3] ✉, Hong Zhu ®[7] ✉ & Lin Hao ®[1] ✉

All-insulator heterostructures with an emerging metallicity are at the forefront of material science, which typically contain at least one band insulator while it is not necessary to be. Here we show emergent phenomena in a series of all-correlated-insulator heterostructures that composed of insulating $CaIrO_3$ and insulating $La_{0.67}Sr_{0.33}MnO_3$. We observed an intriguing insulator-to-metal transition, that depends delicately on the thickness of the iridate component. The simultaneous enhancements of magnetization, electric conductivity, and magnetoresistance effect indicate a percolation-type nature of the insulator-to-metal transition, with the percolation threshold can be reached at an exceptionally low volume fraction of the iridate. Such a drastic transition is induced by an interfacial charge transfer, which interestingly alters the electronic and crystalline structures of the bulk region rather than the limited ultrathin interface. We further showcased the central role of effective correlation in modulating the insulator-to-metal transition, by demonstrating that the critical thickness of iridate for triggering the metallic state can be systematically reduced down to a single unit-cell layer.

Interface is at the center of modern electronics. The celebrated example is a PN junction in semiconductor heterostructures. By replacing traditional semiconductors with transition metal oxide insulators, interface of the obtained heterostructures is unveiled to host even more intriguing physical properties[1,2]. One of the pioneering works is the discovery of a metallic interface in heterostructures composed of insulating $LaTiO_3$ and $SrTiO_3$[3,4]. The subsequent observation of 2D superconductivity in $SrTiO_3/LaAlO_3$ heterostructures further stimulates the condensed matter community to search for electrically conducting interface in various insulating heterostructures[5,6]. The main thread here is sample design by virtue of the alternate positive/negative charged layers in transition metal oxides, such that the polar discontinuity at the interface will give rise to high-mobility electron gas at the 2D limit due to polar catastrophe. Very recently, metallicity and even superconductivity were also observed in insulating heterostructures composed of $KTaO_3$[7–9].

At the interface, the conduction electrons by themselves are strongly correlated because of dimensional confinement[1]. The effective correlation was unveiled to be a key parameter for driving the insulator-to-metal transition at the interface[10]. On the other hand, the

[1]Anhui Key Laboratory of Low-Energy Quantum Materials and Devices, High Magnetic Field Laboratory, HFIPS, Chinese Academy of Sciences, Hefei, Anhui, China. [2]Science Island Branch of Graduate School, University of Science and Technology of China, Hefei, China. [3]National Synchrotron Radiation Laboratory, University of Science and Technology of China, Hefei, China. [4]Information Materials and Intelligent Sensing Laboratory of Anhui Province, Key Laboratory of Structure and Functional Regulation of Hybrid Materials of Ministry of Education, Institutes of Physical Science and Information Technology, Anhui University, Hefei, China. [5]School of Physics and Astronomy, Beijing Normal University, Beijing 100875, China. [6]Department of Physics, Pusan National University, Busan, South Korea. [7]Department of Physics, University of Science and Technology of China, Hefei, China. ✉e-mail: kaichen2021@ustc.edu.cn; zhuh@ustc.edu.cn; haolin@hmfl.ac.cn

main building blocks of the above-mentioned insulating hetero-structures, i.e., SrTiO$_3$ and KTaO$_3$, are both band insulators where the effect of effective correlation is negligible. As such, whether and to what extent the interface conductivity is modulated by the effective correlation of the pristine constituents is in fact an open question. From the material point of view, the heterostructures host at least one band insulator, while all-correlated-insulator based heterostructures are much less explored.

In contrast to band insulators, a correlated insulator itself may also exhibit a rich phase diagram covering both insulating and metallic phases[11]. In order to achieve a conducting interface of an all-correlated-insulators based heterostructure, it is intuitive to explore correlated compounds which are characteristic of a robust metallic ground state but can be manipulated to be insulating. Perovskite iridates appear as an elegant candidate considering the robust (semi-) metallicity due to the well-extended $5d$ orbitals[12–14]. For example, it is predicted that a large effective correlation is essential in order to destabilize the Dirac-type (semi-) metallicity in perovskite CaIrO$_3$ (CIO)[15]. Similarly, the $3d$ La$_{0.67}$Sr$_{0.33}$MnO$_3$ (LSMO) could be another candidate because of the robust half metallicity thanks to the large bandwidth[16]. Recently, it was reported that a large tensile strain stabilizes an insulating state in LSMO due to the reduced bandwidth $W$[17,18] and thus the enhanced effective correlation $U/W$, where $U$ is the on-site Coulomb interaction. In addition, it is typically observed that interfacial effect between metallic iridates and manganites could be exceptionally strong[19–23], which is especially beneficial for promoting phase transitions in the composed heterostructures.

In this work, we constructed a series of CIO/LSMO hetero-structures. Both components were driven to be strong insulators by virtue of a large tensile strain. The heterostructures, however, host variable electronic and magnetic ground states by atomically mod-ulating the thickness of each constituent. The digitally controllable insulator-to-metal transition is unveiled to be driven by the

substantial charge transfer at the Ir/Mn interface, which promotes a ferromagnetic metallic state in LSMO by altering the electronic structure as well as the octahedral rotation pattern. Through a modulation of effective correlation, we obtained an insulator-to-metal transition at the critical thickness of only one-unit cell, ren-dering effective correlation the fundamental as well as an energetic parameter in tailoring emergent phenomena of correlated insulating heterostructures.

## Results and Discussion

### Emergent insulator-to-metal transition in CIO/LSMO heterostructures

Epitaxial strain is an efficient knob for modulating bond angle and bond length, which dominate the bandwidth as well as effective correlation in oxide thin films[24]. Note that CIO is a robust Dirac semimetal[15] while LSMO is a half metal with a large bandwidth[25]. To achieve an insulating state, we adopted the cubic KTaO$_3$ (KTO, $a = 3.988$ Å) substrate in order to introduce a large tensile strain on CIO (pseudo-cubic, $a = 3.868$ Å) as well as LSMO (pseudo-cubic, $a = 3.876$ Å)[26], as schematically shown in Fig. 1a. A series of CIO/LSMO heterostructures were prepared on (001)-oriented substrates using a pulsed laser deposition system equipped with a RHEED unit. The optimized growth condition is included in the Methods. The LSMO thickness is selected to be 20 unit cells, such that the LSMO block is thick enough to avoid the dead layer problem[27]. In the meanwhile, the LSMO block is sufficiently thin to fully sustain the large tensile strain from the substrate, as representatively shown from the reci-procal space mapping (RSM) measurement in Fig. 1b, which is crucial for this study as will be discussed later. Transmission electron microscopy measurements demonstrate that the interface between CIO and LSMO (Fig. 1c), as well as the interface between LSMO and substrate are well-identified see Supplementary note 2. The observed sharp interruptions of the Ir/Mn elements as well

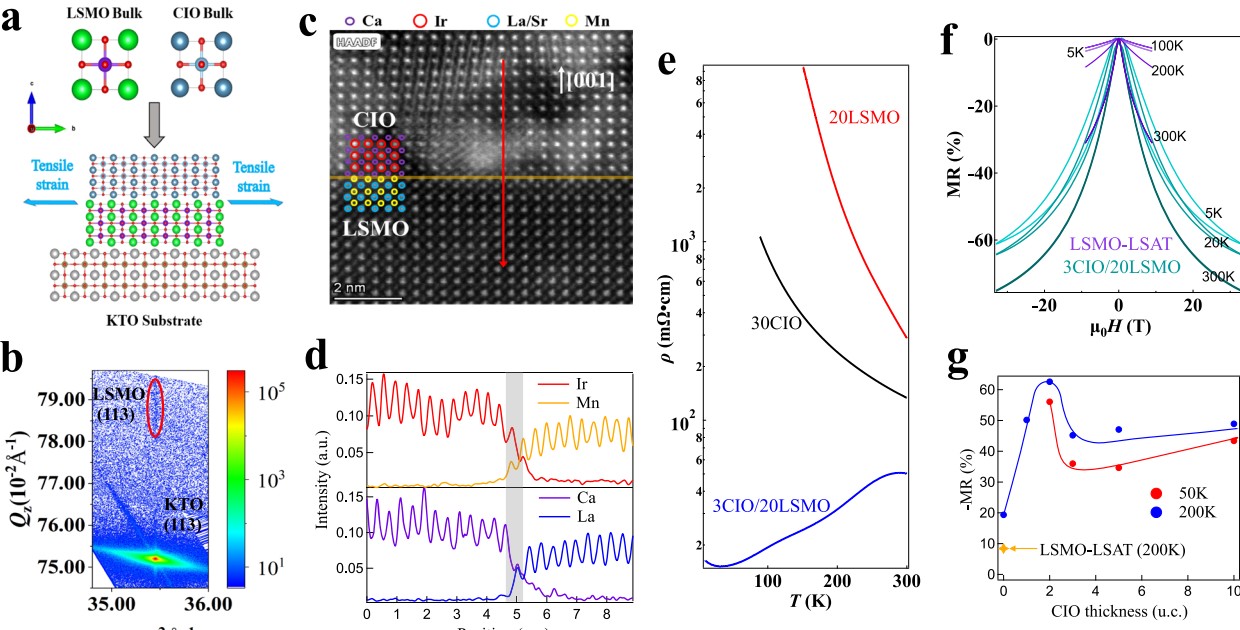

**Fig. 1 | Metallic heterostructures composed of correlated insulators. a** Schematic diagram of CIO/LSMO heterostructures grown on KTO substrates. **b** X-ray reciprocal space mapping measurements of the 3CIO/20LSMO hetero-structure around the (113)-peak of the KTO substrate. **c** Representative high-angle annular dark-field (HAADF) image of a 30CIO/30LSMO heterostructure. The orange line marks the interface between LSMO and CIO. **d** Electron energy loss spectro-scopy (EELS) spectra of $B$-site ions (Ir and Mn) and $A$-site ions (Ca and La) across the

CIO/LSMO interface. The interface region is marked by a grey zone. **e** Temperature dependent resistivity of 20LSMO film, CIO (30-unit cells) film and 3CIO/20LSMO heterostructure. **f** In-plane magnetic field-dependent magnetoresistance (MR) of LSMO-LSAT film and 3CIO/20LSMO heterostructure at various temperatures. **g** CIO thickness dependence of MR at 5 K and 200 K. MR of the LSMO-LSAT film at 200 K is also shown for comparison.

as the Ca/La elements in the EELS profile (Fig. 1d) indicates that chemical intermixing crossing the interface is within ~one unit cell.

The effective correlation induced insulating states were unveiled from Fig. 1e, where both CIO and 20LSMO thin films grown on KTO are strongly insulating with a rapid increase of resistivity upon reducing temperature. The insulating nature of CIO film is in stark contrast to the persistent (semi-) metallicity in CIO films grown on substrates with a smaller lattice parameter[28,29], highlighting the robust (semi-) metallicity in CIO and the crucial role of a strong effective correlation in destroying it. However, when depositing the two insulating constituents sequentially, we observed a significant change in electronic properties in the obtained heterostructure. We hereafter use $n$CIO/$m$LSMO to denote the heterostructure with a CIO block of $n$ unit cells and a LSMO block of $m$ unit cells. Figure 1e is a showcase for the metallic ground state in a 3CIO/20LSMO heterostructure where the resistivity decreases with decreasing temperature. From the magnetoresistance (MR) measurements, it was found that the heterostructure features a large field-dependent hysteresis, with the saturation field even stronger than 30 T at 20 K (Fig. 1f). In addition, MR is stronger at high temperatures in the 3CIO/20LSMO heterostructure while MR is maximized at low temperatures in the 20LSMO film see Supplementary note 10. The difference of the electronic properties between 3CIO/20LSMO and the 20LSMO film was also observed in electrostatic force microscopy measurements see Supplementary note 3. The non-monotonic dependence of the MR on $n$ (Fig. 1g) indicates that large MR of the heterostructure is not an inherent property of the CIO block, either. Indeed, MR of a CIO single crystal diminishes quickly with increasing temperature[15], and MR effect of CIO films is almost unobservable at high temperatures see Supplementary note 10. We then conclude that the heterostructure hosts an emergent electronic state that distinct from the tensile-strain stabilized insulating state in single LSMO or CIO film.

Additionally, we note that the conductivity of the heterostructure is closely analogous to the bulk LSMO with a double-exchange mediated metallic phase. In order to make a direct comparison, we prepared a nearly strain-free LSMO film on a LSAT substrate, which indeed displays a good metallicity as expected see Supplementary note 4. Nevertheless, the bulk-like LSMO-LSAT film displays a negligible hysteresis in the MR measurements (Fig. 1f), in stark contrast to the large MR hysteresis in the 3CIO/20LSMO heterostructure. Furthermore, the MR magnitude of the 3CIO/20LSMO heterostructure is substantial even at temperatures as low as 5 K, while MR of the LSMO-LSAT film is negligible at low temperatures. Moreover, as compared to the saturation field in magnetization measurements see Supplementary note 5, the saturation field in MR of the heterostructure is stronger by more than two orders of magnitude. This large difference implies that traditional MR mechanisms, such as scattering on magnetic domain walls, play a negligible role in the MR, which otherwise will lead to the same coercivity in MR and magnetization. We note that a large MR hysteresis was also observed in charge ordered manganites due to the field induced melting of the charge ordered state[25]. This scenario, however, is also unlikely because the heterostructure is metallic rather than insulating. Furthermore, the melting of a charge ordered state would lead to 4~5 orders of magnitude reduction in resistivity[25], which is much more pronounced than the MR in Fig. 1f. In addition to above scenarios, a large MR hysteresis is frequently believed to be the hallmark of phase separation in manganites[16,30,31]. Here, while magnetization of the ferromagnetic clusters is easily saturated under magnetic field, the electronic conducting paths continue to grow even under a strong magnetic field leading to the progressive reduction in electric resistivity. The large MR hysteresis accounts for the dissimilar evolution processes of the electronic conducting paths upon increasing and decreasing magnetic field. The magnetotransport measurements thus suggest that the heterostructure presents substantial phase separation in the manganite block, in contrary to the single ferromagnetic phase in the strain-free bulk-like film.

## Percolation nature of the insulator-to-metal transition

To shed additional light on the emergent electronic state, we systematically varied the CIO block thickness from 1 to 10-unit cells in the heterostructures. As shown in Fig. 2a, while the 1CIO/20LSMO heterostructure remains an insulator, the insulating state is clearly reduced as compared to the 20LSMO film. In presence of a CIO bilayer, the 2CIO/20LSMO heterostructure displays a peculiar electronic property, where the resistivity is almost independent of temperature. Continuing increase of the CIO thickness leads to an emergent metallic state in the 3CIO/20LSMO heterostructure, as mentioned in the last section. The electric conductivity can be hardly improved with further increasing the CIO thickness. For instance, the electric resistivity of 5CIO/20LSMO and 10CIO/20LSMO is only slightly different from that of the 3CIO/20LSMO heterostructure. The ceased improvement of electric conductivity suggests that an interfacial effect governs the insulator-to-metal transition in the heterostructures.

In complementary to the electric transport measurements that are sensitive to the conducting portions, magnetic measurements probe the entire sample. Figure 2b shows the temperature dependent magnetization of the heterostructures with various CIO block thicknesses. In the insulating 20LSMO film (i.e., $n = 0$), a broad magnetic transition can be identified around 260 K. In comparison to the profound ferromagnetic metallic LSMO-LSAT film see Supplementary note 4, we note that both the onset temperature and saturated magnetization are significantly reduced, in consistent with previous reports[17]. Interestingly, onset temperature is clearly enhanced when combining with a single unit cell of CIO. Even more striking is the about-doubled magnetization in the 1CIO/20LSMO as compared to the pristine 20LSMO film. The magnetization continues to increase with the CIO block thickness until reaching the maximum in the 3CIO/20LSMO. The independence of magnetization on the CIO block thickness when $n$ is larger than 3 indicates that it is the manganite block dominates the overall magnetic properties. Indeed, the parent compound CIO is paramagnetic with no observable magnetic transition[15]. In fact, almost all the perovskite iridates crystallize into a paramagnetic state or a spin canted antiferromagnetic state with a weak magnetization[32–34]. Magnetization from iridates is similarly small even in artificially designed heterostructures. For instance, the spontaneous magnetization is less than 0.1 $\mu_B$/Ir in iridate/titanite as well as iridate/cobaltite heterostructures[35–40], and the net magnetizations are about 0.04 $\mu_B$/Ir and 2 $\mu_B$/Mn in SrIrO$_3$/manganite heterostructures[19,20,41]. Therefore, we adopted the volume of the LSMO block rather than the overall heterostructure to calculate the magnetization in Fig. 2b.

The modulation of physical properties can also be uncovered from magnetotransport measurements. Knowing that the temperature $T_{MR}$ that maximizes the MR effect is a crucial parameter in manganites[42], we plotted $\rho(0)/\rho(0.5\,T)$ that normalized by the maximum in Fig. 2c in order to better illustrate the $n$-dependent $T_{MR}$. For the insulating 20LSMO film, the MR effect under a 0.5 T field should be maximized at certain temperatures below 150 K. The insulating as well as the metallic heterostructures are characteristic of a broad hump in the temperature dependence of MR. $T_{MR}$ increases drastically with the CIO block thickness first and then approaches the maximum in the 3CIO/20LSMO. While MR of the metallic LSMO-LSAT film also features a nonmonotonic temperature dependence, the peak is significantly sharper than those in the heterostructures. By extracting the magnetic ordering temperature $T_C$ from the inflection

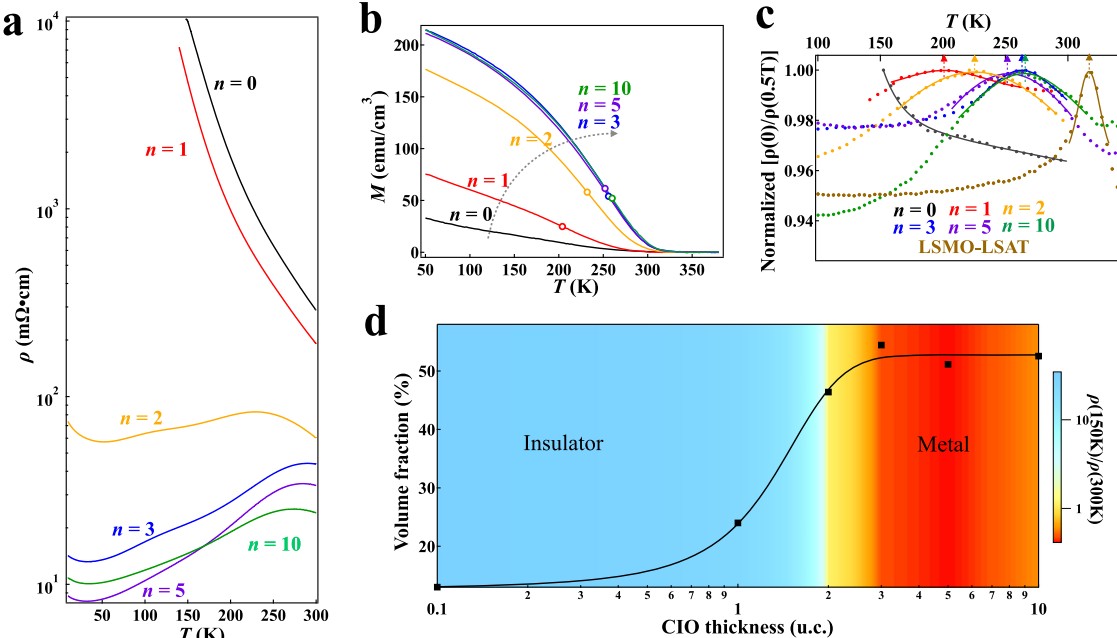

**Fig. 2 | Strongly $n$-dependent electronic and magnetic states. a** Temperature-dependent resistivity of the $n$CIO/20LSMO heterostructures. The resistivity of each heterostructure was calculated by only taking into consideration of the thickness of the LSMO block. **b** In-plane magnetization measured under a 0.1 T field as a function of temperature. The magnetization was estimated with the volume of the LSMO block (See main text for detailed discussion). The hollow circles indicate the inflection points of the heterostructures. **c** Temperature dependence of $\rho(0)/$ $\rho(0.5\,\text{T})$ normalized by the maximum for the $n$CIO/20LSMO heterostructures. Data of the LSMO-LSAT films is also shown for comparison. The solid lines are guides to the eye. The dashed arrows denote the temperatures $T_{\text{MR}}$ that maximize $\rho(0)/$ $\rho(0.5\,\text{T})$. **d** CIO thickness dependence of the volume fraction of the ferromagnetic metallic clusters. The black line is a guideline. The colorful background is extrapolated from $\rho(150\,\text{K})/\rho(300\,\text{K})$ of various $n$CIO/20LSMO heterostructures.

point of the temperature dependent magnetization, it is clear that $T_{\text{MR}}$ is essentially the same as $T_C$ in the LSMO-LSAT film. This is a well-established correlation between MR and magnetization in double-exchange manganites[42]. Similarly, $T_{\text{MR}}$ is also consistent with $T_C$ in the heterostructures see Supplementary note 7, further demonstrating that a manganite-like ferromagnetic order gradually develops and eventually dominates the overall magnetization in the heterostructure. Nonetheless, the broad $T$-dependent bump indicates that the heterostructures are more complex than the profound double-exchange ferromagnetic materials. Indeed, a broad transition is typically observed in phase-separated manganites[16,43], where certain magnetic moments are not fully aligned. The observation that $T_C$ of the heterostructures is always lower than that of the LSMO-LSAT film further suggests presence of spin disorder in the former.

The scenario of phase separation is essentially of disconnected ferromagnetic and metallic clusters embedding in an antiferromagnetic and insulating matrix[16]. With the clusters growing progressively, a conducting path emerges at the percolation threshold and dominates the overall conductivity[16,25]. In order to better characterize the insulator-to-metal transition, we explored the insulating strength by extracting the relative increase of resistivity from room temperature to low temperature, $\rho(150\,\text{K})/$ $\rho(300\,\text{K})$. As shown in Fig. 2d, the insulating strength well describes the $n$-dependent insulator-to-metal transition in a colorful illustration. In addition, the phase separation scenario can also be quantified from evolution of the volume fraction of the ferromagnetic metallic clusters ($V_{\text{FM}}$). Firstly, we assume that the bulk-like LSMO-LSAT film is fully magnetized with the entire sample volume being a single ferromagnetic phase, i.e., $V_{\text{FM}} = 1$. In the phase-separated heterostructures, it is safe to assume a negligible magnetization in the insulating regions and a substantial magnetization that equals

to that of the LSMO-LSAT in ferromagnetic clusters. Along this line, $V_{\text{FM}}$ is essentially the ratio between the saturation magnetization of the heterostructure and that of the LSMO-LSAT film. In this context, $V_{\text{FM}}$ was extracted and summarized in Fig. 2d. $V_{\text{FM}}$ increases drastically with the CIO block thickness and reaches the saturated value ~52% when the heterostructures enter into the metallic region. Figure 2d also demonstrates that 2CIO/20LSMO is at the boundary of the insulator-to-metal transition. This state is special where the size of ferromagnetic clusters may be optimized, such that the clusters can be easily connected under magnetic fields while the field-induced resistivity change is still substantial considering the non-metallic state under zero field, leading to the maximized MR in Fig. 1g. We note that the saturated value is remarkably consistent with the classical percolation threshold of 50%, confirming a percolation-driven nature of the ferromagnetic metallic state[16]. In conventional solid solution systems where percolation is driven by chemical intermixing, the percolation percentage is essentially the substitution/doping ratio. Taking the 1CIO/20LSMO heterostructure for example, however, it is striking that $V_{\text{FM}}$ is increased by more than 20% although the volume fraction of the CIO block is only ~5%. This comparison explicitly excludes the possibility of a percolation process driven by the chemical intermixing of Ir/Mn, but rather renders the heterostructure a novel system with an exceptionally high percolation efficiency.

## Electronic and structural origins of the transition
To uncover the driven force of the efficient percolation process, we performed X-ray absorption (XAS) as well as synchrotron XRD measurements. As shown in Fig. 3a, we observed a profound Mn $L$-edge absorption profile, with mixed $Mn^{3+}$ and $Mn^{4+}$, in all the heterostructures as well as the 20LSMO reference sample. For the

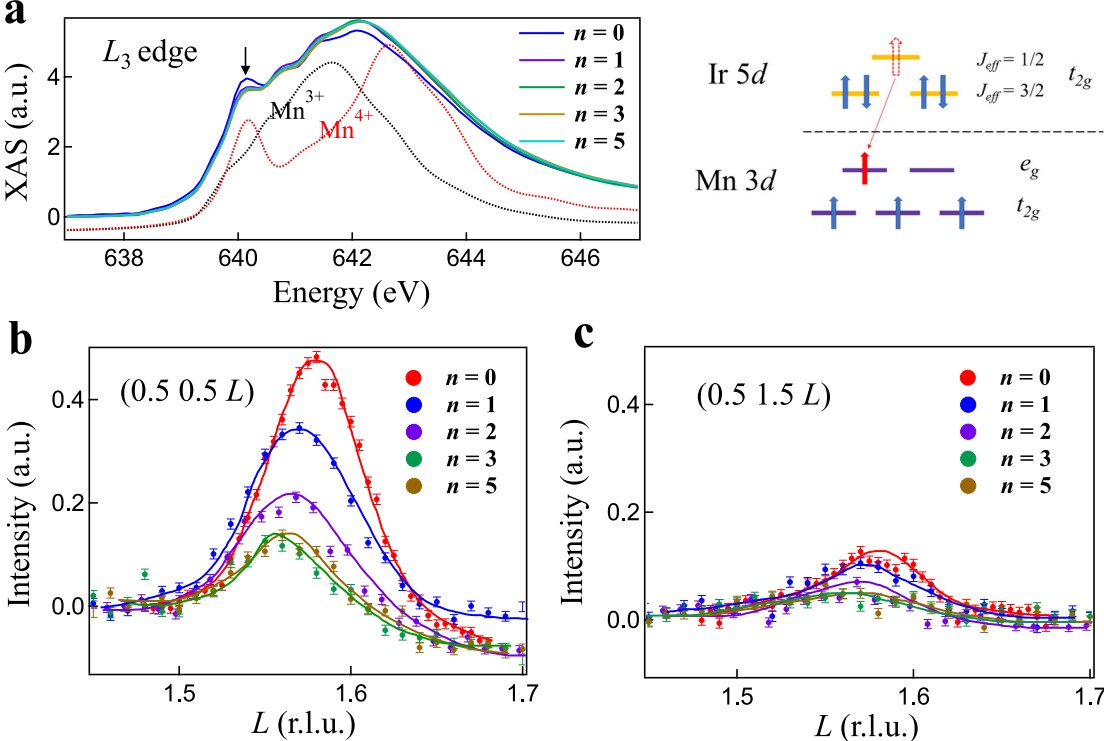

**Fig. 3 | Systematic variations of valence state and octahedral rotation pattern.**
**a** XAS spectra at the Mn $L_3$-edge of the $n$CIO/20LSMO heterostructures. Spectra from reference samples with only Mn$^{3+}$ (LaMnO₃) or Mn$^{4+}$ (Li₂MnO₃) are also shown for comparison[44]. Arrow denotes the feature of Mn$^{4+}$ in the spectra. Right panel is the schematic illustration of electron transfer from the Ir-5$d$ orbitals to the Mn-3$d$ orbitals. **b**, **c** Room-temperature synchrotron XRD around the (0.5 0.5 1.5) and (0.5 1.5 1.5) Bragg reflections, respectively. The reciprocal space is defined based on the crystal structure of KTO. Error bar represents the statistic error.

pristine 20LSMO film, we observed a sharp lower-energy peak at around 640.3 eV at the Mn $L_3$-edge. As a comparison, the lower-energy peak is significantly reduced in the 1CIO/20LSMO heterostructure. Since the lower-energy peak is characteristic of the Mn$^{4+}$ ions[44], the observation indicates that content of Mn$^{3+}$ was increased, i.e., extra electrons were introduced in the LSMO block after interfacing with one unit cell of CIO. The estimated charge transfer is about 0.1 electron per perovskite unit. Similarly, we note that charge transfer as large as 0.5 electron per perovskite unit cell was reported in SrIrO₃/SrMnO₃ and CaIrO₃/CaMnO₃ heterostructures[19,45]. The reduced charge transfer in our heterostructures may be related to the partial occupancy of $e_g$ orbitals in LSMO, while the tetravalent manganites are characteristic of unoccupied $e_g$ orbitals[19,45]. The large tensile strain may also play a role in the reduced charge transfer due to the strain-shifted $e_g$ orbitals in manganites[22]. In contrast to the observed large variation in the 1CIO/20LSMO heterostructure, further increasing the thickness of CIO block leads to a negligible modulation in the XAS profile. Since the soft XAS only probes the surface region, the almost unchanged XAS profiles in the $n$CIO/20LSMO ($n \geq 1$) heterostructures indicate that the Mn$^{4+}$-to-Mn$^{3+}$ transition is easily saturated especially for the top couple layers. The charge transfer thus promotes the ferromagnetic metallic state electronically, because of the fact that extra electrons expands the regime of ferromagnetic phase while destabilizing the antiferromagnetic state of LSMO even under a large tensile strain[46]. Along this line, we expect that similar insulator-to-metal transition may not be observable in heterostructures composed of the insulating LSMO and a Mott insulator if there is no substantial charge transfer effect, such as the LSMO/LaTiO₃ heterostructures[47,48].

Showing in Fig. 3b is the Bragg peak that probes the octahedral tilting around the diagonal direction of the $ab$-plane. The peak intensity is gradually suppressed with increasing the CIO block thickness when $n \leq 3$. Note that the peak intensity is proportional to the squared of octahedral tilting[49,50], this observation thus explicitly points out that the out-of-plane Mn-O-Mn bond angle has been systematically enhanced in the CIO/LSMO heterostructures as compared to the 20LSMO film. The almost unchanged peak intensity in the 3CIO/20LSMO and 5CIO/20LSMO heterostructures further confirms that an interfacial effect dominates the insulator-to-metal transition in the heterostructures. However, since the XRD signal is dominated by the overall crystal structure, in contrast the surface-sensitive probe of XAS, the systematic variation of XRD peak intensity when $n \leq 3$ unveiled that the interfacial effect alters the lattice structure of the entire LSMO block rather than the limited interfacial region. The enhanced out-of-plane bond angle also gives rise to an increase in the out-of-plane lattice constant, as can be seen from the low-angle peak shift with $n$ in Fig. 3b.

We note that the 20LSMO film features a strong tilting peak in Fig. 3b, unveiling a small out-of-plane bond angle that deviates significantly from 180°. This is understandable because the tensile strain shrinks the out-of-plane lattice parameter by reducing the out-of-plane bond angle. On the other hand, the in-plane bond angle is enlarged due to the expanded in-plane lattice, as inferred from the weak rotation peak in Fig. 3c. Considering the fact that a straight Mn-O-Mn bond is especially beneficial for the double exchange process, the insulating nature of the pristine 20LSMO film then must be dictated by the small out-of-plane bond angle. The improved electric conductivity in heterostructures thus can be apprehended by the enhanced out-of-plane angle with increasing $n$. Interestingly, we note that the already-weakened octahedral

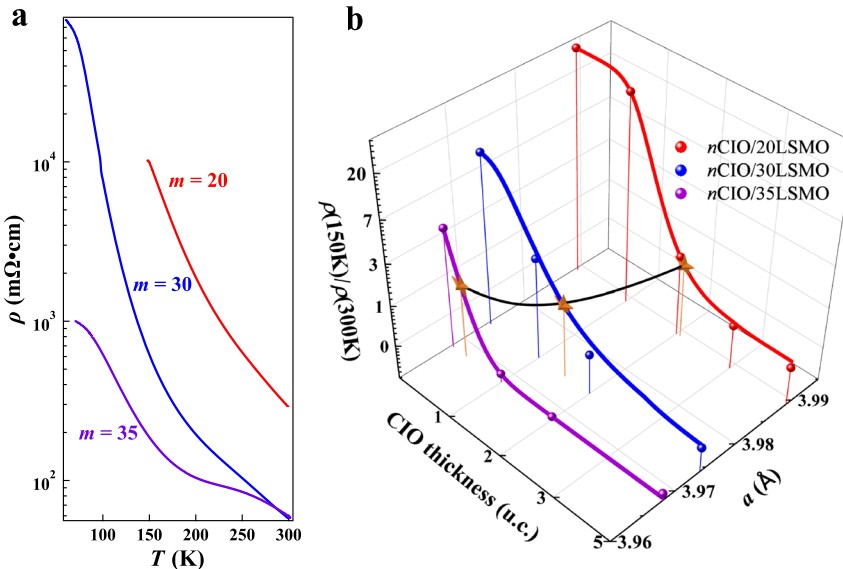

**Fig. 4 | Modulation of effective correlation. a** Temperature-dependent resistivity of the LSMO (20, 30, 35 u.c.) films. **b** Evolution of the insulating strength $\rho(150\,K)/\rho(300\,K)$ with CIO thickness and in-plane lattice parameter $a$. Stars represent the critical $n$ values at which $\rho(150\,K)/\rho(300\,K) = 1$, and the black curve represents the variation of critical $n$ with the in-plane lattice parameter.

rotation is also suppressed due to the interfacial effect (Fig. 3c), which should further facilitate the double exchange process. Along this line, the ferromagnetic state is favored in the CIO/LSMO heterostructures from the structural point of view. The cooperative interplay of the electronic and structural modulation renders the charge transfer an efficient knob not only in modulating the electronic state but also the microscopic lattice structures of quantum materials beyond the limited interfacial region.

**Accelerating the transition by tailoring effective correlation**

Having identified the central role of electronic/structural effect in the emergent insulator-to-metal transition, we next explore potential routes to boost the intriguing transition. Here, driving the insulating LSMO closer to the metallic state is the key. We would like to highlight that the strong tensile strain or the large effective correlation is essential in stabilizing the insulating state of LSMO[46,51], and epitaxial strain typically relaxes with increasing film thickness. We then systematically increased the thickness of the LSMO block in order to gradually reduce the tensile strain. Reducing the tensile strain essentially leads to an enhanced bandwidth[18] and a reduced effective correlation. Three series of heterostructures with the LSMO block of 20-, 30-, and 35-unit cells were prepared. In each series, the CIO block was varied from 1- to 10-unit cells. As summarized in Fig. 4a, the thick LSMO films host a reduced electric resistivity, demonstrating a destabilized insulating state. The decreased in-plane lattice parameter in Fig. 4b confirms that the effective correlation is reduced thanks to the partially relaxed tensile strain.

We found this variation has a profound impact on the critical thickness of the CIO block in triggering the emergent insulator-to-metal transition. With the reduced effective correlation in the $n$CIO/30LSMO heterostructures, we observed a similar $n$-dependent insulator-to-metal transition as the $n$CIO/20LSMO heterostructures but at a reduced critical $n$ of 2 (Fig. 4b). In the percolation scenario, the reduced insulating strength in the pristine LSMO films (Fig. 4a) can be phenomenologically ascribed to the shortened separation between metallic clusters. The enlarged metallic clusters thus are easier to be connected with the interfacial effect, promoting the insulator-to-metal transition in the heterostructures. The reduced critical $n$ with increasing LSMO thickness also rules out possible contribution from Sr-segregation, which would otherwise give rise to an opposite trend due to escalated Sr-segregation in thick LSMO films[52]. With the further reduced effective correlation due to enhanced strain relaxation in the $n$CIO/35LSMO heterostructures, we found the critical thickness is further reduced and the insulator-to-metal transition occurs even when CIO is only of one-unit cell thick. Therefore, the above observation demonstrates that heterostructures composed of correlated insulators not only represent as a fertile playground for exploring emergent phenomena but also host a great application potential upon delicate engineering.

To summarize, we have constructed a series of heterostructures composed of CIO and LSMO on KTO substrates. Both constituents are characteristic of a robust metallic ground state in the bulk phase, but were turned to be strong insulators due to a substantially large tensile strain. In contrast to the two insulating constituents, the heterostructures display an emergent insulator-to-metal transition when digitally varying the thickness of CIO block. Through combined electronic and structural characterizations, we discovered an intriguing percolation-like insulator-to-metal transition. The efficient percolation transition was unveiled to be driven by an intriguing Ir/Mn interfacial effect, which alters both electronic and crystalline structures of the manganite block in the bulk region rather than the ultrathin interface region. The effective correlation was unveiled to be an essential parameter in the emergent insulator-to-metal transition, such that the critical thickness can be tailored to a single unit cell. This work renders the interface between two correlated insulators a versatile and flexible platform for discovering and engineering emergent phenomena.

## Methods

### Sample synthesis

$n$CaIrO$_3$/$m$La$_{0.67}$Sr$_{0.33}$MO$_3$ ($n$CIO/$m$LSMO) ($n$ = 0, 1, 2, 3, 5, 10 u.c., $m$ = 20, 30, 35 u.c.) heterostructures were deposited on (001)-oriented KTaO$_3$ substrates by pulsed laser deposition (PLD) techniques. A reflection high-energy electron diffraction (RHEED) unit is utilized to monitor the growth. During deposition, O$_2$ gas was introduced at a constant partial pressure of 75 mTorr, and the growth temperature was maintained at 750 °C for all the films. PLD was carried out using a KrF excimer UV laser ($\lambda$ = 248 nm), for which the energy density and

repetition rate were set to 1.5 J/cm² and 2 Hz, respectively. After deposition, the films were cooled to room temperature at a rate of 10 °C/min with the oxygen partial pressure of about 350 Torr.

## Characterization techniques

The X-ray diffraction patterns were measured on a Panalytical X'Pert MRD diffractometer. Electric transport properties were investigated on a physical properties measurement system (Quantum Design). Magnetic properties were measured on a vibrating sample magnetometer (Quantum Design). High-field MR measurement was performed on the Steady High Magnetic Field Facilities (WM5), High Magnetic Field Laboratory, Chinese Academy of Sciences. The X-ray absorption spectroscopy measurements were performed using linearly polarized X-ray at BL07U beamline of Shanghai Synchrotron Radiation Facility (SSRF), using a total electron yield (TEY) detection method. Synchrotron X-ray diffraction experiments were performed at the BL02U2 beamline in SSRF. The electrostatic force microscopy option is implemented on commercial SPM systems (Nanoscope-V Multimode AFM) with the tapping-lift mode. To detect the electrostatic forces, a voltage (+4 V) was applied to AFM tips coated with Pt metal with a tip-sample separation of 50 nm. The cross-sectional electron transparent samples for TEM observations were fabricated by a Carl Zeiss crossbeam 550 L FIB-SEM using the conventional lift-out method operating at 30 kV following a similar procedure in Refs. 53,54. The aberration-corrected scanning transmission electron microscopy (STEM), and energy dispersive X-ray spectroscopy (EDS) experiments were carried out at 300 kV using a Thermo Fisher Scientific Themis Z microscope.

## Data availability

All data needed to evaluate the conclusions in the paper are present in the paper and/or the Supplementary Materials. Additional data related to this paper may be requested from the authors.

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

## Acknowledgements

We appreciate helpful discussions with S. Dong, J.X. Zhang, and Z. Liao. We also appreciate prompt experimental assistances from X. Zhang and H.F. Du. This work was supported by the Collaborative Innovation Program of Hefei Science Center, Chinese Academy of Sciences (2022HSC-CIP005), the international partnership program of the Chinese Academy of Sciences (145GJHZ2022044MI), the HFIPS Director's Fund (2023YZGH01), the High Magnetic Field Laboratory of Anhui Province (Grant No. AHHM-FX-2021-03), the National Natural Science Foundation of China (11874359 and 12104460), the Anhui Provincial Major S&T Project (s202305a12020005) and the Basic Research Program of the Chinese Academy of Sciences Based on Major Scientific Infrastructures (JZHKYPT-2021-08). The authors would like to appreciate BL02U2 and BL07U Beamlines at SSRF for the synchrotron beamtime. A portion of this work was performed on the Steady High Magnetic Field Facilities, High Magnetic Field Laboratory, Chinese Academy of Sciences.

## Author contributions

L.H. conceived and directed the study. E.M. undertook sample growth. E.M. and D.L. performed in-house structural characterization. E.M., D.L., H.Z., Z.Q. and J.C. performed synchrotron XRD measurements. L.W. and K.C. performed XAS measurements. E.M., Z.W. and C.X. performed high-field transport measurements. D.S. performed TEM characterizations. Y.L. conducted EFM measurements. H.J., E.M., K.C., H.Z. and L.H. analyzed data. E.M. and L.H. wrote the manuscript with inputs from all other coauthors.

## Competing interests

The authors declare no competing interests.
