## [Peer Review File · Nature Communications]

An atomically controlled insulator-to-metal transition in iridate/manganite heterostructuresReviewers' Comments:

Reviewer #1:

Remarks to the Author:

Men et al. presented an investigation into the insulator-to-metal transition in heterostructures composed of insulating CaIrO_3 and $\text{La}_{0.67}\text{Sr}_{0.33}\text{MnO}_3$. The authors have observed a transition from an insulating state to a metallic state, possibly interface related, which depends on the thickness of the iridate component. They found that the transition is driven by a cooperative interplay of electronic variation and structural modification at the interface between CaIrO_3 and $\text{La}_{0.67}\text{Sr}_{0.33}\text{MnO}_3$. The authors further argued the role of electronic correlation in the emergent phenomena of this all insulator heterostructure, which may provide insights into the mechanisms behind such a transition. I think this is an interesting observation and a clear set of data. While the observed 'recovery' of a correlated metallic state for interfacial manganite layer can be taken as an interesting emergent phenomenon, the current form of the data presentation and the discussions on the possible implications of the results do not stand as new observations/discoveries nor in-depth understanding of the underlying physics. Therefore, I cannot recommend publication of the current form of the manuscript. I suggest the authors to improve their ways of presentation and explanation, to fully reflect in a revised version what are the genuinely key findings and what are the possible mechanisms, perhaps by adding more data and reformulating the discussion part. My questions/comments are appended below, which may hopefully be helpful for improving the manuscript.

1. A general comment: the authors seem to have put a significant emphasis on 'correlated' or 'strongly-correlated'. While these two components are indeed 'correlated oxides', the key findings to this study do not clearly show any particular linkage to correlation effect. I was a bit confused on this point.
2. Related to Point 1, I had the impression that the authors would like to appeal the readers by stating that this is an all-correlated-insulator heterostructures. However, even if this is true, it does not mean that correlation plays a role in the occurrence of such a metallic state. If correlation does play an important role driving phenomena shown in the work, experimental evidence that directly reveals correlation effects should be presented. Currently, there seems lack of such information. In fact, to me, the key finding here is a ferromagnetic metallic state found between two paramagnetic and/or antiferromagnetic (or weak ferromagnetic) layers. This could be due to charge transfer effect, as discussed by the authors, or reduced charge depletion (or dead layer) enabled by a cap layer, etc., but has no clear relation to the electronic correlation. I strongly suggest the authors to re-visit this issue across the entire manuscript.
3. Due to the criticisms above, the authors may consider revising the title. One example could be: An emerging correlated (or ferromagnetic) metallic state at atomically controlled iridate/manganite heterostructures.
4. Line 53-55: the statement 'the interfacial superconductivity also displays similar features with the unconventional superconductivity in strongly correlated cuprates' may not be a correct or up-to-date statement. Suggest to remove or revise.
5. Line 62-63: the authors mentioned '5d orbitals are well-extended, and electronic correlation is typically insufficient to stabilize an insulating ground state'; however, in their experiments, the CaIrO_3 shows a clear insulating behavior. The authors should comment on this, which serves an important

reasoning why an insulating CIO was used.

6. Also related, Line 67-68: the authors mentioned that they carefully designed the CIO layer to be insulating by tuning correlation. In fact, such an insulating behavior may just directly originate from the strain effect or the substrate induced band-width change. The authors should clarify this point, either by choosing the words more carefully or providing more useful references to support their arguments.

7. Line 71-72: Upon reading through, I still cannot apprehend what the authors mean by 'promote a ferromagnetic metallic state...structurally'. Do the authors mean strain induced lattice effect?

8. How about using other Mott systems, such as LaTiO_3 , instead of CIO? Since they can also provide charges, can authors predict if it would work?

9. Line 89-90: this statement is simply not correct. The reference cited here (Ref. 15) only discusses the role of epitaxial strain on ferroic materials, such as stabilizing ferroic orders that do not exist in unstrained parent compounds; it does not discuss how electronic correlation can be tuned by strain. In general, the strain-tuned correlation strength shows a complex behavior.

10. Line 91-92: again, 'correlation induced insulating state' is not an accurate statement.

11. Line 103: I am not sure how the authors reached the conclusion of 'negligible chemical intermixing', as only a TEM image of a cropped small interface area was shown, without the needed discussion on the EELS data (too small to see in the inset). I would like to request that the authors to show relatively large area TEM images, covering both interfaces and ideally also the surface of the heterostructure, and properly discuss the results obtained by EELS, which will give key information on the quality of the interface.

12. Line 109: the word 'alternating' is not accurate as the heterostructure is just a bilayer structure.

13. Line 112: the field-dependent hysteresis could be due to a robust ferromagnetic state. No detailed analysis was done (ideally which should be in conjunction with Hall data). The discussions related to this point did not provide a clear interpretation of hysteresis.

14. Line 117-118: there should be no 'surprise' on the observation that the heterostructure is more conducting than CIO single-crystals, as one is measuring a much more conducting ferromagnetic metallic layer of manganite. So I think the authors are comparing apple and orange here.

15. Line 127: Evidence should be provided to support the claim 'strong-correlation-stabilized'.

16. Line 140: this observation (independent of the thickness of CIO when larger than 3 uc) can be taken as an evidence that this is an interface effect. I suggest to revise this sentence.

17. Line 160: from the references provided by the authors, it appears that there were other iridate/manganite heterostructures that were constructed and reported before. The authors should make proper reference and discussions to these literature in the introduction and motivation part.

18. In Figure 2c (and the discussion on this panel), I did not catch why using a normalized resistivity with respect to ρ (0.5T). The authors should explain/define more clearly.

19. Line 176: the statement 'should be maximized at temperatures below 100 K' is just an inferred statement, not a scientific statement. Should be removed or rewritten.

20. Line 193: I do not see why this confirms the presence of spin frustration.

21. The authors should describe more in detail how they extract V_{FM} , the volum fraction. Also, the magnetization data as well as V_{FM} for the reference sample (LSMO on LSAT) should also be provided.

22. Line 230: why the level of charge transfer (0.1 e per uc) is smaller than that (0.5 e per uc) in other similar heterostructures reported in the literature? Can authors elaborate?

23. Line 254-256: I don't understand this statement. Shouldn't it be the other way around (i.e. wouldn't this further corroborate interface being the key?)?
24. Line 259-260: if this is true (that both the o-o-p and i-p bond angles are driven close to 180 deg in the heterostructures, and this is the reason for the appearance of a metallic state), why LSMO on KTO substrates under large tensile strain would show insulating behavior? Tensile strain should straighten the bond and increase the bond angle.
25. Line 264-265: the statement 'charge transfer an efficient knob in switching the antiferromagnetic insulating state to the ferromagnetic metallic state' does not represent a new phenomenon, as this has been observed many times in previous literature.
26. Line 270-272: why changing the thickness of CIO can change the correlation? The authors should discuss more.
27. Line 288: I don't really understand the statement here 'as the enlarged ferromagnetic clusters due to the reduced electronic correlation'.
28. Line 298: why mentioning 'a Dirac semimetal', if no such aspect has been discussed or demonstrated, nor it be relevant to the findings here.
29. In the supplementary materials, the AFM images should be updated with images of higher quality.

Reviewer #2:

Remarks to the Author:

The manuscript report insulator-to-metal transition in iridate (CIO)/manganite (LSMO) heterostructures. Individual thin films show insulating properties, while CIO/LSMO heterostructure show metallic properties. In addition, the structures show higher MR ratios than the LSMO layer, which is another interesting finding. Basically, the manuscript is of high interest for oxide people. However, there are some issues that should be clarified with revision.

1. For insulator-to-metal transitions in $ABO_3/SrTiO_3$ -based heterostructures, A site cations tend to make basic oxides. This means that the elements have strong activity to oxidizing. In this study, Ca is a strong oxidizer, which takes oxygen from LSMO to make oxygen vacancies in LSMO. With oxygen vacancies, electron concentration in LSMO can increase. In this aspect, the insulator-to-metal transition happens with carrier doping. How do you think about this?
2. Related with the above comment, oxygen spectra in XPS and XAS should be checked carefully.
3. Density-functional theory calculation is missing in the manuscript, which is crucial to understand the mechanism.
4. In my opinion, other $CaMeO_3$ compounds can bring insulator-to-metal transition in LSMO. Could you try this to see the results?
5. How about the stability of the heterostructures in ambient oxygen pressure annealing?

Reviewer #3:

Remarks to the Author:

Reviewer #4:

Remarks to the Author:

The paper “Atomically controlled insulator-to-metal transition in strongly correlated iridate/manganite heterostructures” by Men et al. reports a study on $\text{La}_{0.67}\text{Sr}_{0.33}\text{MnO}_3$ thin films capped with different thicknesses (n in unit cells) of CaIrO_3 (CIO/LSMO). The resistivity, magnetization, XAS and synchrotron XRD measurements were done varying n from 0 to 5 unit cells. They report that the electronic transport evolves from insulating to metallic behavior when $n > 2$. The Curie temperature and magnetization also increase at around this thickness. The authors concluded that the CIO thickness in CIO/LSMO heterostructure is critical to the observed phenomena, driven by a Mn^{4+} -to- Mn^{3+} charge transfer at the interface. XRD symmetric scans are reported in the SI for some of the main samples of the studies. Reciprocal space mapping and STEM imaging is reported for different samples in Figure 1.

The authors claim novelty based on their observed metallicity in a heterostructure consisting only of correlated insulators. LSMO and CIO are both metals in their groundstate and can be made insulating by biaxial strain, e.g. on KTaO_3 . The authors do mention this but do not adequately rule out that their heterostructure is allowing one or more of the constituent materials to return to their bulk-like properties through a trivial effect such as strain relaxation. Other major concerns that could trivially lead to a similar result are CIO-capping effect on LSMO quality, strain-induced critical CIO thickness, and CIO/LSMO heterostructure quality based on STEM.

In its current form we do not find the manuscript acceptable for publication. It may be acceptable after addressing our below concerns, approximately in order of importance:

1. Sample characterization is lacking.

- a. The STEM image in Figure 1c is too limited. Please provide a low magnification image and an image including KTO/LSMO interface (as is written in the text) and the sample surface.
- b. Why are there intensity modulations and some diagonal stripes in the CIO layer?
- c. Provide a STEM image for one of the main samples in the study, $n=2$ or 3 for instance. The one currently shown is $n=30$ which, according to Figure 4 is relaxed. We suspect CIO does not grow so well on fully strained LSMO.
- d. Interfacial intermixing is very likely occurring. Especially for $n=1, 2$, and given the high energy of the PLD growth parameters, it is unlikely that the interface is sharp, or even distinguishable for these low thicknesses. STEM-EELS is essential in ruling out interfacial intermixing, please provide that in addition to the HAADF images.
- e. The reciprocal space map in Figure 1b is only of a LSMO film, not one of the bilayers studied in the rest of the paper. An RSM including CIO is essential in ruling out strain relaxation in the CIO cap. Here we accept a thicker CIO layer because probably $n < 5$ will not be visible.

2. The electrostatic force microscopy images (Supplementary Figure 2) look like they are dominated by scanning artifacts. E.g. the horizontal streaks.

a. The repeated structures that are light on the right and dark on the left suggest that the signal is not real – in the reverse trace does the image look the same?

b. We think the images mostly reflect a very rough surface topography. Is it possible to provide just AFM topography image and quantify roughness?

3. The authors should rule out a trivial capping effect.

a. This scenario seems possible based on the EFM images in SI Figure 2. Both images have similar bumps so they are probably originating in the LSMO. But the right image has 3 u.c. of CIO on top and the surface does look a little smoother. It therefore seems likely that the CIO capping layer is acting to preserve the LSMO quality. Possibly suppressing strontium migration to the surface. The authors should comment.

b. Have they ever tried an “inert” capping layer that shouldn’t induce charge transfer? LaAlO_3 or SrTiO_3 for instance? If there really is physics driving the observed capping layer thickness-dependent behavior then this would be an easy way to show it.

4. The authors say that the difference in MR between their capped LSMO and the “bulk” LSMO is what proves that they have an “emergent” metallicity and not just bulk-like metallicity. In particular they see a large MR at low T with 3 u.c. of CIO and a small MR when “bulk-like”. This is probably the most important data of the paper so they should show the CIO thickness-dependence. Looking at the resistivity, $n=2$ should be measurable at least. The MR behavior might be coming from CIO, this should rule it out.

5. The XRD data:

a. The symmetric XRD scans shown in SI Figure 1: Why is the CIO peak not visible? $n=10$ should certainly be thick enough. Are the authors not concerned that their CIO is not crystalline?

b. Looks like the c lattice parameter is increasing with n. This could be due to a gradual relaxation of the tensile strain. As mentioned above, RMS of the whole series are needed to rule this out. Otherwise it looks like the LSMO is returning to its bulklike properties.

c. Also, the LSMO peak intensity diminishes with increasing CIO. Could this also explain the reduction in intensity, with increasing cap thickness, of the half-order peaks shown in Figure 3b-c?

d. In Figure 3b-c it looks like there are fits to the data. Are these fits, and if so, what are the fit parameters? The FWHM would be especially important to show as evidence that the structural distortion is constant over the whole film, and constant with increasing n.

6. The findings of Figure 4, that a partially relaxed LSMO needs less CIO cap in order to become metallic, seems to support our suspicion that strain (relaxation) of one or both of the layers play a larger role than interfacial charge transfer. Have the authors not tried thinner LSMO?

7. If we extrapolate Figure 4 to lower a lattice parameter then we would conclude that this “emergent” metallic state of LSMO should be stabilized with no capping layer, is this the case?

8. The authors make volume fraction arguments based on the low temperature magnetization, noting that the 1 u.c. CIO cap increases M by 20% while the volume fraction of CIO is only 5%. As the authors have not provided evidence that there is no cation intermixing and the high energy growth probably leads to 2-3 unit cells of intermixing, the volume fraction of the interface could be $3/18 = 17\%$. This matches quite well with the magnetic volume fraction determination. The authors need to address this possibility.

9. Would significant hysteresis be expected in the resistivity versus temperature if there is a

percolation mechanism at play?

10. More information is needed on how charge transfer was estimated from the XAS. Details on peak fitting, parameters, goodness of fit and uncertainty on the charge transfer value itself should be provided.

11. Do the authors have any comment on the kinks at ~120K and ~200K in Fig. 1d and Fig. 2a metallic curves?

12. What thickness do the authors assume when calculating the resistivity, that all the LSMO becomes metallic and nothing else?

13. SI Figure S4 makes the saturation regime clear but not the coercive field, please show a close-up of the low field data.

14. Can the authors add arrows to Figure 2b, similar to 2c, showing the inflection points that they discuss in the text?

15. In Fig. 2d, could they add mention if the line is a guide to eye or something else?

16. Please add the field direction to Figure 1e-f.

17. Remove the word “alternating” from the 2nd paragraph of “results and discussion” – it implies a repeating superlattice structure but these samples are just bilayers.

Reviewer #5:

Remarks to the Author:

Report of the First Referee

Men et al. presented an investigation into the insulator-to-metal transition in heterostructures composed of insulating CaIrO_3 and $\text{La}_{0.67}\text{Sr}_{0.33}\text{MnO}_3$. The authors have observed a transition from an insulating state to a metallic state, possibly interface related, which depends on the thickness of the iridate component. They found that the transition is driven by a cooperative interplay of electronic variation and structural modification at the interface between CaIrO_3 and $\text{La}_{0.67}\text{Sr}_{0.33}\text{MnO}_3$. The authors further argued the role of electronic correlation in the emergent phenomena of this all insulator heterostructure, which may provide insights into the mechanisms behind such a transition. I think this is an interesting observation and a clear set of data. While the observed ‘recovery’ of a correlated metallic state for interfacial manganite layer can be taken as an interesting emergent phenomenon, the current form of the data presentation and the discussions on the possible implications of the results do not stand as new observations/discoveries nor in-depth understanding of the underlying physics. Therefore, I cannot recommend publication of the current form of the manuscript. I suggest the authors to improve their ways of presentation and explanation, to fully reflect in a revised version what are the genuinely key findings and what are the possible mechanisms, perhaps by adding more data and reformulating the discussion part. My questions/comments are appended below, which may hopefully be helpful for improving the manuscript.

Response:

Thank you for the instructive comments. We sincerely appreciate your efforts in reviewing our work. We would like to highlight that we discovered a novel percolation-type insulator-to-metal transition in insulating iridate/manganite heterostructures, thanks to the interfacial charge transfer which not only modulates the electronic state but also the crystalline structure of the bulk region rather than the limited interface region. Next, we list detailed response to each comment.

1. A general comment: the authors seem to have put a significant emphasis on ‘correlated’ or ‘strongly-correlated’. While these two components are indeed ‘correlated oxides’, the key findings to this study do not clearly show any particular linkage to correlation effect. I was a bit confused on this point.

Response:

This comment is crucial. We thank you for raising this point. We should clarify that the interfacial effect is the driving force for the insulator-to-metal transition in the heterostructures, while effective electronic correlation is essential for stabilizing an insulating state in the parent compounds. We would like to explain it in more detail.

The motivation is to realize an insulator-to-metal transition in heterostructures composed of two correlated insulators. Towards this end, it is intuitive to find two correlated oxides, which are insulating but have a strong tendency towards the metallic ground state. Therefore, no matter which component (or both components) is driven to be a metal by the interfacial effect, the heterostructure will have an insulator-to-metal transition. LSMO directly comes into mind because of the well-known robust metallic

ground state due to the large bandwidth. CIO has a similar robust metallic (semi-metallic) ground state, but rather due to the Dirac-type band structure. The robust metallic ground states, on the other hand, indicate that an exceptionally large stimulus is needed for stabilizing an insulating state in both materials. Indeed, it is predicted that a strong correlation is necessary in order to transition the CIO into an insulator [*Nat. Commun.* 10, 362 (2019)]. It is also difficult to turn LSMO into an insulator, but it was recently reported that LSMO/KTO is an insulator due to the reduced bandwidth W under a large tensile strain [*Nano Lett.* 22, 7066 (2022)]. The reduced bandwidth W gives rise to an enhanced effective correlation U/W , where U is the coulomb interaction. Along this way, one can see that a strong effective correlation is essential to stabilize the insulating phases in the two parent compounds, which is the precondition for preparing the all-correlated-insulator-based heterostructures.

In order to give a clear motivation, we add the following part in the introduction. “In contrast to band insulators, a correlated insulator itself may also exhibit a rich phase diagram covering both insulating and metallic phases¹¹. In order to achieve a conducting interface of an all-correlated-insulators based heterostructure, it is intuitive to explore correlated compounds which are characteristic of a robust metallic ground state but can be manipulated to be insulating. Perovskite iridates appear as an elegant candidate considering the robust (semi-) metallicity due to the well-extended $5d$ orbitals¹²⁻¹⁴. For example, it is predicted that a large effective correlation is essential in order to destabilize the Dirac-type (semi-) metallicity in perovskite CaIrO_3 (CIO)¹⁵. Similarly, the $3d$ $\text{La}_{0.67}\text{Sr}_{0.33}\text{MnO}_3$ (LSMO) could be another candidate because of the

robust half metallicity thanks to the large bandwidth¹⁶. Recently, it was reported that a large tensile strain stabilizes an insulating state in LSMO due to the reduced bandwidth W ^{17,18} and thus the enhanced effective correlation U/W , where U is the on-site Coulomb interaction.” on Page 4.

The role of effective correlation is also manifested in the last portion of this work. We have identified that the insulator-to-metal transition is because of the fact that the interfacial effect triggers an intriguing metallic state in the LSMO block. In order to facilitate the insulator-to-metal transition, the next step is to further destabilize the insulating state of the LSMO block. Effective correlation is an efficient knob for this purpose, in the sense that a reduced effective correlation will destabilize the insulating phase of manganites [*Low Temp. Phys.* 26, 171 (2000)].

In order to better explain the role of effective correlation in the sample design, we add a statement in Section 4, “Here, driving the insulating LSMO closer to the metallic state is the key. We would like to highlight that the strong tensile strain or the large effective correlation is essential in stabilizing the insulating state of LSMO^{48,53}, and epitaxial strain typically relaxes with increasing film thickness, we then systematically increased the thickness of the LSMO block in order to gradually reduce the tensile strain. Reducing the tensile strain essentially leads to an enhanced bandwidth¹⁸ and a reduced effective correlation.”.

2. Related to Point 1, I had the impression that the authors would like to appeal the readers by stating that this is an all-correlated-insulator heterostructures. However, even if this is true, it does not mean that correlation plays a role in the occurrence of such a

metallic state. If correlation does play an important role driving phenomena shown in the work, experimental evidence that directly reveals correlation effects should be presented. Currently, there seems lack of such information. In fact, to me, the key finding here is a ferromagnetic metallic state found between two paramagnetic and/or antiferromagnetic (or weak ferromagnetic) layers. This could be due to charge transfer effect, as discussed by the authors, or reduced charge depletion (or dead layer) enabled by a cap layer, etc., but has no clear relation to the electronic correlation. I strongly suggest the authors to re-visit this issue across the entire manuscript.

Response:

Thank you for this comment. We realized that we should elucidate the relation between tensile strain and effective correlation well in the introduction. Please refer the last response, where we explain the role of a large effective correlation (i.e., a strong tensile strain) in our work.

In particular, in the last portion of this work, we prepared heterostructures with LSMO blocks of 20, 30, and 35 unit cells in order to tailor the effective correlation. With increasing LSMO thickness, the tensile strain is partially relaxed as confirmed by the slightly reduced in-plane lattice parameter [supplemenray Fig. 8]. The relaxed tensile strain indicates that the bandwidth W of LSMO is increased, giving rise to a reduced effective correlation U/W as well as the insulating strength in Fig. 4a. Eventually, the partially relaxed LSMO is closer to its metallic phase, and an insulator-to-metal transition is achieved in the 1CIO/35LSMO heterostructure even the CIO cap

is only of one-unit cell thick. Here, effective correlation U/W plays a role in reducing the threshold of CIO thickness that triggers the insulator-to-metal transition in the heterostructures.

3. Due to the criticisms above, the authors may consider revising the title. One example could be: An emerging correlated (or ferromagnetic) metallic state at atomically controlled iridate/manganite heterostructures.

Response:

Thank you for this suggestion, we would like to change the title to “An atomically controlled insulator-to-metal transition in iridate/manganite heterostructures”, considering your last comments on our improper usage of “correlation” and the intriguing observation of the atomically controlled insulator-to-metal transition.

4. Line 53-55: the statement ‘the interfacial superconductivity also displays similar features with the unconventional superconductivity in strongly correlated cuprates’ may not be a correct or up-to-date statement. Suggest to remove or revise.

Response:

Thank you for the suggestion, this statement is removed.

5. Line 62-63: the authors mentioned ‘5d orbitals are well-extended, and electronic correlation is typically insufficient to stabilize an insulating ground state’; however, in their experiments, the CIO shows a clear insulating behavior. The authors should

comment on this, which serves an important reasoning why an insulating CIO was used.

Response:

CIO is a Dirac semimetal, and it was predicted that a strong effective correlation is necessary to stabilize an insulating state [*Nat. Commun.* 10, 362 (2019)]. In perovskite oxides, the bandwidth is dominated by $W \propto \cos \varphi/d^{3.5}$, where d is the bond length and φ is the buckling angle. During epitaxial growth, epitaxial strain alters the bond length as well as buckling angle. As a result, the bandwidth W also varies with epitaxial strain. We agree with the referee that the relation between bandwidth and epitaxial strain is not straightforward. Nonetheless, the tunable bandwidth W paves the way for an efficient manipulation of effective correlation U/W in thin films, which is actually a more relevant parameter (as compared to Hubbard U) in experimental condensed matter physics.

Experimentally, CIO films were successfully grown on a series of substrates. However, the reported CIO films preserve the semimetallic nature [See for example, *Appl. Phys. Lett.* 107, 012104 (2015); *J. Appl. Phys.* 117, 195305 (2015)]. The previous works demonstrate the robustness of the semimetallic state in CIO. In this work, the large tensile strain from KTO leads to a significant reduction of bandwidth. As a return, the enhanced effective correlation is sufficiently strong to stabilize the insulating state in CIO.

We add a discussion “The insulating nature of CIO film is in stark contrast to the persistent (semi) metallicity in CIO films grown on substrates with a smaller lattice

parameter^{29,30}, highlighting the robust (semi) metallicity in CIO and the crucial role of a strong effective correlation in destroying it.” on Page 6 in the revised manuscript.

6. Also related, Line 67-68: the authors mentioned that they carefully designed the CIO layer to be insulating by tuning correlation. In fact, such an insulating behavior may just directly originate from the strain effect or the substrate induced band-width change. The authors should clarify this point, either by choosing the words more carefully or providing more useful references to support their arguments.

Response:

According to your suggestion, we revised the statement as “Both components were driven to be strong insulators by virtue of a large tensile strain.” on Page 4 in the updated manuscript.

7. Line 71-72: Upon reading through, I still cannot apprehend what the authors mean by ‘promote a ferromagnetic metallic state...structurally’. Do the authors mean strain induced lattice effect?

Response:

In Figs. 3b and 3c, we show results on the microscopic lattice structure. Specifically, octahedral tilting is significantly suppressed in the heterostructures. The observation demonstrates that the interfacial effect not only changes the electronic structure, but also changes the crystal structure. The enlarged bond angles, as indicated

from the reduced octahedral tilting/rotation, are beneficial for stabilizing the ferromagnetic metallic state even without the modulation of the electronic state.

To be more specific, we revise this statement as “which promotes a ferromagnetic metallic state in LSMO by altering the electronic structure as well as the octahedral rotation pattern.” on Page 4 in the updated manuscript.

8. How about using other Mott systems, such as LaTiO_3 , instead of CIO? Since they can also provide charges, can authors predict if it would work?

Response:

Thank you for this suggestion. The driving force of the insulator-to-metal transition is the cooperative electronic and structural variation due to the interfacial effect between LSMO and CIO. We note that similar interfacial effect, such as charge transfer, was not observed in a LSMO/LTO/STO heterostructure [*Nat. Commun.* 13, 5631 (2022)]. Furthermore, a recent work also suggests that charge transfer between LSMO and LTO is unlikely to occur [*Phys. Rev. B* 100, 115119 (2019)]. From the above literatures, we think replacing CIO with LTO in the heterostructures may not be able to trigger an insulator-to-metal transition.

We add a statement on this prediction “Along this line, we expect that similar insulator-to-metal transition may not be observable in heterostructures composed of the insulating LSMO and a Mott insulator if there is no substantial charge transfer effect, such as the LSMO/ LaTiO_3 heterostructures^{49,50}.” on Page 14. The references are also cited properly.

9. Line 89-90: this statement is simply not correct. The reference cited here (Ref. 15) only discusses the role of epitaxial strain on ferroic materials, such as stabilizing ferroic orders that do not exist in unstrained parent compounds; it does not discuss how electronic correlation can be tuned by strain. In general, the strain-tuned correlation strength shows a complex behavior.

Response:

We agree with you that the relation between effective correlation and epitaxial strain is not straightforward. We revised it as “Epitaxial strain is an efficient knob for modulating bond angle and bond length, which dominate the bandwidth as well as effective correlation in oxide thin films” on Page 4 to be more accurate. The reference is replaced with [*Adv. Mater.* 23, 3363 (2011)].

10. Line 91-92: again, ‘correlation induced insulating state’ is not an accurate statement.

Response:

Thank you for this suggestion, and we revised it as “an insulating state” on Page 6 in the updated manuscript.

11. Line 103: I am not sure how the authors reached the conclusion of ‘negligible chemical intermixing’, as only a TEM image of a cropped small interface area was shown, without the needed discussion on the EELS data (too small to see in the inset).

I would like to request that the authors to show relatively large area TEM images,

covering both interfaces and ideally also the surface of the heterostructure, and properly discuss the results obtained by EELS, which will give key information on the quality of the interface.

Response:

Thank you for this comment. A TEM image of a large area is provided in Fig. R1. In order to demonstrate the quality of interface, we also present the enlarged view of the CIO-LSMO interface as well as the LSMO-KTO interface on the right panels. Both interfaces are sharp. The surface of the heterostructure, while is also well identified, is slightly damaged during TEM sample preparation. In fact, TEM measurement on iridate sample is a well-known challenging task, because Ir element easily losses during FIB preparation. We now include the large-area image in the supplementary materials.

Figure R1| A large-area TEM image of the 30CIO/30LSMO heterostructure.

Figure R2 is the EELS data presented in the inset of Fig. 1c. Relative intensities of Ir and Mn elements (*B*-site) crossing the CIO/LSMO interface are shown. For better comparison, we also present the relative intensities of Ca and La elements (*A*-site). The observed sharp interruptions of the Ir/Mn elements as well as the Ca/La elements indicate that the chemical intermixing crossing the interfaces are within one unit cell. We properly discuss the EELS data in the updated manuscript by adding “The observed sharp interruptions of the Ir/Mn elements as well as the Ca/La elements in the EELS profile (Fig. 1d) indicates that chemical intermixing crossing the interfaces are within ~one unit cell.” on Page 6 to address the interface quality. The EELS data is shown in a separate panel in the updated Fig. 1 for a better presentation.

Figure R2| The updated EELS data of the 30CIO/30LSMO heterostructure. The grey zone indicates the interface region.

12. Line 109: the word ‘alternating’ is not accurate as the heterostructure is just a bilayer structure.

Response:

Yes, you are right. We removed “alternating” in the statement.

13. Line 112: the field-dependent hysteresis could be due to a robust ferromagnetic state. No detailed analysis was done (ideally which should be in conjunction with Hall data). The discussions related to this point did not provide a clear interpretation of hysteresis.

Response:

The MR hysteresis indeed indicates presence of a ferromagnetic order in the heterostructure. We have also performed Hall measurements. Unfortunately, we did not obtain a reliable Hall data set because of the dominated MR effect in the Hall channel.

In the updated manuscript, we add a discussion on the possible mechanisms of the large hysteresis. On Page 8, we add “Moreover, as compared to the saturation field in magnetization measurements²⁷, the saturation field in MR of the heterostructure is stronger by more than two orders of magnitude. This large difference implies that traditional MR mechanisms, such as scattering on magnetic domain walls, play a negligible role in the MR, which otherwise will lead to the same coercivity in MR and magnetization. We note that a large MR hysteresis was also observed in charge ordered manganites due to the field induced melting of the charge ordered state²⁵. This scenario, however, is also unlikely because the heterostructure is metallic rather than insulating.

Furthermore, the melting of a charge ordered state would lead to 4~5 orders of magnitude reduction in resistivity²⁵, which is much more pronounced than the MR in Fig. 1f. In addition to above scenarios, a large MR hysteresis is frequently believed to be the hallmark of phase separation in manganites^{16,31,32}. Here, while magnetization of the ferromagnetic clusters is easily saturated under magnetic field, the electronic conducting paths continue to grow even under a strong magnetic field leading to the progressive reduction in electric resistivity. The large MR hysteresis accounts for the dissimilar evolution processes of the electronic conducting paths upon increasing and decreasing magnetic field. The magnetotransport measurements thus suggest that the heterostructure may present substantial phase separation in the manganite block, in contrary to the single ferromagnetic phase in the strain-free bulk-like film.”.

14. Line 117-118: there should be no ‘surprise’ on the observation that the heterostructure is more conducting than CIO single-crystals, as one is measuring a much more conducting ferromagnetic metallic layer of manganite. So I think the authors are comparing apple and orange here.

Response:

Thank you for this comment. We would like to point out that both CIO and LSMO have a robust metallic ground state, and it is in fact not straightforward to pinpoint which part dominates the conductivity of the whole heterostructure without the subsequent magnetic and MR measurements. Similar to LSMO, there is also a chance that CIO is recovered to the metallic phase. Therefore, we think it is reasonable to

compare the heterostructure with both constituents in an unbiased way at the early stage of this research. Nevertheless, in order to avoid any confusion, we revised it as “Additionally, we note that the conductivity of the heterostructure is closely analogous to the bulk LSMO with a double-exchange mediated metallic phase.” on Page 7.

15. Line 127: Evidence should be provided to support the claim ‘strong-correlation-stabilized’.

Response:

We revised the statement as “We then conclude that the heterostructure hosts an emergent electronic state that distinct from the tensile-strain stabilized insulating state in single LSMO or CIO film.” on Page 7.

16. Line 140: this observation (independent of the thickness of CIO when larger than 3 uc) can be taken as an evidence that this is an interface effect. I suggest to revise this sentence.

Response:

Thank you for this point, we revised it as “The ceased improvement of electric conductivity suggests that an interfacial effect governs the insulator-to-metal transition in the heterostructures.” on Page 9.

17. Line 160: from the references provided by the authors, it appears that there were other iridate/manganite heterostructures that were constructed and reported before. The

authors should make proper reference and discussions to these literature in the introduction and motivation part.

Response:

The reported large charge transfer at Ir/Mn interface in metallic iridate/manganite heterostructure is one of the motivations for designing the heterostructures. We add “In addition, it is typically observed that interfacial effect between metallic iridates and manganites could be exceptionally strong¹⁹⁻²³, which is especially beneficial for promoting phase transitions in the composed heterostructures.” on Page 4 in the introduction part.

We would also like to highlight that our work is distinct from the previous works because manganite and/or iridate is metallic in the relevant references [*Nat. Commun.* 7, 12721 (2016); *PNAS* 113, 6397-6402 (2016); *Phys. Rev. Lett.* 119, 077201 (2017); *Nano Lett.* 17, 2126-2130 (2017); *Adv. Mater.* 33, 2008269 (2021)], which are now properly cited.

18. In Figure 2c (and the discussion on this panel), I did not catch why using a normalized resistivity with respect to $\rho(0.5T)$. The authors should explain/define more clearly.

Response:

The ratio between zero-field resistivity and resistivity at 0.5 T, $\rho(0)/\rho(0.5T)$, is inversely proportional to the magnetoresistance, $[\rho(0.5T)-\rho(0)]/\rho(0)$. In

double-exchange manganites, magnetoresistance maximizes around the magnetic transition temperature T_c . The similarity between T_c and the temperature T_{MR} that maximizes the MR effect is another evidence that MR of the heterostructure is dominated by the LSMO component. In other words, T_{MR} is more relevant than the MR magnitude in this study. To better illustrate the variation of T_{MR} with n , we thus plotted $\rho(0)/\rho(0.5T)$ that normalized by the maximum value in Fig. 2c.

In the updated manuscript, we add “Knowing that the temperature T_{MR} that maximizes the MR effect is a crucial parameter in manganites⁴⁴, we plotted $\rho(0)/\rho(0.5T)$ that normalized by the maximum in Fig. 2c in order to better illustrate the n -dependent T_{MR} .” on Page 11 to explain the definition.

19. Line 176: the statement ‘should be maximized at temperatures below 100 K’ is just an inferred statement, not a scientific statement. Should be removed or rewritten.

Response:

Thank you for pointing it out. This is a typo. We revised it as “should be maximized at certain temperatures below 150 K” on Page 11.

20. Line 193: I do not see why this confirms the presence of spin frustration.

Response:

Thank you for this comment. The suppression of T_c is an indication of destabilization of the long-range magnetic order, i.e., presence of spin disorder. To be

more accurate, we revised the statement as “further suggests presence of spin disorder in the former” on Page 12.

21. The authors should describe more in detail how they extract V_{FM} , the volume fraction. Also, the magnetization data as well as V_{FM} for the reference sample (LSMO on LSAT) should also be provided.

Response:

Thank you for this suggestion. We assumed that the LSMO-LSAT film is a single ferromagnetic phase, i.e., $V_{FM} = 1$, considering the bulk-like nature. The magnetization data of the LSMO-LSAT film was provided in the supplementary materials.

In order to provide more detail in deducing V_{FM} of heterostructure, we revised “*Here, magnetization of the bulk-like LSMO-LSAT film sets the upper bound for the spontaneous magnetization in the ferromagnetic clusters. For the residual antiferromagnetic matrix, it is safe to assume a negligible magnetization.*” as “Firstly, we assume that the bulk-like LSMO-LSAT film is fully magnetized with the entire sample volume being a single ferromagnetic phase, i.e., $V_{FM} = 1$. In the phase-separated heterostructures, it is safe to assume a negligible magnetization in the insulating regions and a substantial magnetization that equals to that of the LSMO-LSAT in ferromagnetic clusters. Along this line, V_{FM} is essentially the ratio between the saturation magnetization of the heterostructure and that of the LSMO-LSAT film.” on Page 12.

22. Line 230: why the level of charge transfer (0.1 e per uc) is smaller than that (0.5 e

per uc) in other similar heterostructures reported in the literature? Can authors elaborate?

Response:

In tetravalent CaMnO_3 and SrMnO_3 , the e_g orbitals are unoccupied. The amount of e_g holes is maximized. In this case, it might be easy for manganites to accept electrons, giving rise to a large charge transfer. Therefore, the reduced charge transfer is probably related to the reduced amount of e_g holes in LSMO. Tensile strain may also play a role in the reduced charge transfer because of the high-energy shift of e_g orbitals [*Nano Lett.* 17, 2126 (2017)].

We add a statement “The reduced charge transfer in our heterostructures may be related to the partial occupancy of e_g orbitals in LSMO, while the tetravalent manganites are characteristic of unoccupied e_g orbitals^{19,46}. The large tensile strain may also play a role in the reduced charge transfer due to the strain-shifted e_g orbitals in manganites²².” on Page 14 in the revised manuscript to elaborate this point.

23. Line 254-256: I don't understand this statement. Shouldn't it be the other way around (i.e. wouldn't this further corroborate interface being the key?)?

Response:

We would like to clarify that XAS is a surface-sensitive technique while XRD probes the overall crystal structure. In other words, because of the limited volume of interface (around 2 u.c.), the XRD peak intensity is dominated by the entire sample. Therefore, the systematic variation of XRD peaks indicates that the interfacial effect

changes the lattice structure of the entire sample. And, indeed, the almost unchanged intensity in the 3CIO/20LSMO and 5CIO/20LSMO indicates that the interfacial effect is the driving force.

To avoid confusion, we rewrite this statement as “The almost unchanged peak intensity in the 3CIO/20LSMO and 5CIO/20LSMO heterostructures indicates that an interfacial effect dominates the insulator-to-metal transition in the heterostructures. However, since the XRD signal is dominated by the overall crystal structure, in contrast to the surface-sensitive probe of XAS, the systematic variation of XRD peak intensity when $n \leq 3$ unveiled that the interfacial effect alters the lattice structure of the entire LSMO block rather than the limited interfacial region.” on Page 16.

24. Line 259-260: if this is true (that both the o-o-p and i-p bond angles are driven close to 180 deg in the heterostructures, and this is the reason for the appearance of a metallic state), why LSMO on KTO substrates under large tensile strain would show insulating behavior? Tensile strain should straighten the bond and increase the bond angle.

Response:

This is a very good point.

The o-o-p bond angle is significantly reduced while the i-p bond angle is enhanced due to the tensile strain, as inferred from the strong tilting peak of LSMO/KTO in Fig. 3b and the weak rotation peak in Fig. 3c. Please note that, in order to better illustrate the intensity difference, we updated Fig. 3b and Fig. 3c in the same scale with the subtracted background. Since the i-p angle is close to 180° and thus is beneficial for the

double exchange process, the insulating behavior of LSMO/KTO should be rather dominated by the deviation of the o-o-p angle from 180° . This scenario also explains the metallicity in LSMO/LSAT, which has a small i-p bond angle but a large o-o-p bond angle. Therefore, we claim that the enlarged o-o-p bond angle under the interfacial effect dominates the insulator-to-metal transition, while the enhancement of the i-p bond angle plays a secondary and cooperative role.

In the updated manuscript, we revised the “*This observation indicates that both the out-of-plane and in-plane bond angle are driven close to 180° in the heterostructures. A straight Mn-O-Mn bond is especially beneficial for exchanging electrons between Mn^{3+} and Mn^{4+} , and thus facilitates the double exchange process.*” as “We note that the 20LSMO film features a strong tilting peak in Fig. 3b, unveiling a small out-of-plane bond angle that deviates significantly from 180° . This is understandable because the tensile strain shrinks the out-of-plane lattice parameter by reducing the out-of-plane bond angle. On the other hand, the in-plane bond angle is enlarged due to the expanded in-plane lattice, as inferred from the weak rotation peak in Fig. 3c. Considering the fact that a straight Mn-O-Mn bond is especially beneficial for the double exchange process, the insulating nature of the pristine 20LSMO film then must be dictated by the small out-of-plane bond angle. The improved electric conductivity in heterostructures thus can be apprehended by the enhanced out-of-plane angle with increasing n . Interestingly, we note that the already-weakened octahedral rotation is also suppressed due to the interfacial effect (Fig. 3c), which should further facilitate the double exchange process.” on Page 16.

25. Line 264-265: the statement 'charge transfer an efficient knob in switching the antiferromagnetic insulating state to the ferromagnetic metallic state' does not represent a new phenomenon, as this has been observed many times in previous literature.

Response:

We agree that the role of charge transfer has been reported many times previously. In the current work, however, we found that the interfacial charge transfer is novel because it not only plays a role in the interfacial region but also in the region away from the interface. The charge transfer also leads to a cooperative structural effect, which is another rarely explored effect in previous literatures.

In order to highlight this novelty, we would like to revise the statement as "The cooperative interplay of the electronic and structural modulation renders the charge transfer an efficient knob not only in modulating the electronic state but also the microscopic lattice structures of quantum materials beyond the limited interfacial region" on Page 16 in the updated manuscript.

26. Line 270-272: why changing the thickness of CIO can change the correlation? The authors should discuss more.

Response:

We would like to clarify that we prepared a series of $n\text{CIO}/m\text{LSMO}$ heterostructures. The thickness of the LSMO block is changed in order to obtain a partially relaxed tensile strain in LSMO. The bandwidth of LSMO is then will be

enhanced, giving rise to reduced effective correlation U/W in the LSMO block. In each series, the variation of CIO thickness is the same.

To avoid confusion, we revised “*Since electronic correlation is essential in stabilizing the insulating ground states of the parent phases, we then systematically modulated the heterostructure thickness in order to control the electronic correlation.*” to “We would like to highlight that the strong tensile strain or the large effective correlation is essential in stabilizing the insulating state of LSMO^{47,52}, and epitaxial strain typically relaxes with increasing film thickness, we then systematically increased the thickness of the LSMO block in order to gradually reduce the tensile strain. Reducing the tensile strain essentially leads to an enhanced bandwidth¹⁸ and a reduced effective correlation.” on Page 17.

27. Line 288: I don't really understand the statement here ‘as the enlarged ferromagnetic clusters due to the reduced electronic correlation’.

Response:

In the language of epitaxial strain, the enhanced conductivity of pristine LSMO films is due to the partially relaxed tensile strain, which gives rise to a reduced effective correlation. In the percolation scenario, the enhanced conductivity can be ascribed to the shortened separation of metallic clusters, as schematically shown in Fig. R3. As a result, we claim that metallic clusters are enlarged in the partially relaxed LSMO block due to the reduced effective correlation. In this context, it is easier to connect these clusters with the interfacial effect, leading to a complete conducting path in the

heterostructure at a smaller n .

We would like to refer Fig. R3 for a schematic illustration on the manipulated insulator-to-metal transition by taking advantage of variable effective correlation in LSMO.

Figure R3 | The schematical diagram of the percolation-type insulator-to-metal transition. The conductive path is highlighted by red curves.

To avoid potential confusion, we revise the statement as “In the percolation scenario, the reduced insulating strength in the pristine LSMO films (Fig. 4a) can be phenomenologically ascribed to the shortened separation between metallic clusters. The enlarged metallic clusters thus are easier to be connected with the interfacial effect, promoting the insulator-to-metal transition in the heterostructures.” on Page 18 in the updated manuscript.

28. Line 298: why mentioning ‘a Dirac semimetal’, if no such aspect has been discussed

or demonstrated, nor it be relevant to the findings here.

Response:

This statement is related to our motivation, where we planned to find two insulating correlated oxides that have a strong tendency to transition into a metallic state.

To avoid confusion, we revised relevant statement as “To summarize, we have constructed a series of heterostructures composed of CIO and LSMO on KTO substrates. Both constituents are characteristic of a robust metallic ground state in the bulk phase, but were turned to be strong insulators due to a substantially large tensile strain.” on Page 19 in the updated conclusion.

29. In the supplementary materials, the AFM images should be updated with images of higher quality.

Response:

Thank you for this point. We updated the supplementary materials with a series of AFM images for n CIO/20LSMO heterostructures as well as the LSMO-LSAT film. As shown in Fig. R4, all the heterostructures display a similar surface roughness as the LSMO-LSAT film.

Figure R4 | Atomic force microscopy images of n CIO/20LSMO heterostructures and LSMO-LSAT film.

Report of the Second Referee

The manuscript reports insulator-to-metal transition in iridate (CIO)/manganite (LSMO) heterostructures. Individual thin films show insulating properties, while CIO/LSMO heterostructure shows metallic properties. In addition, the structures show higher MR ratios than the LSMO layer, which is another interesting finding. Basically, the manuscript is of high interest for oxide people. However, there are some issues that should be clarified with revision.

Response:

Thank you very much for your positive evaluation and we appreciate all the constructive comments, which we will reponse one-to-one in the following.

1. For insulator-to-metal transitions in $ABO_3/SrTiO_3$ -based heterostructures, A site cations tend to make basic oxides. This means that the elements have strong activity to oxidizing. In this study, Ca is a strong oxidizer, which takes oxygen from LSMO to make oxygen vacancies in LSMO. With oxygen vacancies, electron concentration in LSMO can increase. In this aspect, the insulator-to-metal transition happen with carrier doping. How do you think about this?

Response:

Thank you for this point. Firstly, we would like to refer the updated EELS data (Fig. R2) to elucidate that potential Ca/La intermixing is only around one unit cell in our heterostructures. We thus expect a negligible role of Ca oxidizing in the insulator-to-

metal transition.

Figure R2| The updated EELS data of the 30CIO/20LSMO heterostructure. The grey zone indicates the interface region, which is around one unit cell.

Furthermore, during the epitaxial growth at high temperatures under a high oxygen pressure, it is rather difficult to have an active Ca metal on the surface of LSMO. If in any case, a little amount of Ca ions penetrates into the LSMO block, we suspect that the outcome would be CaMnO_3 , rather than oxygen-deficient LSMO. CaMnO_3 is much more insulating than LSMO and therefore the sample conductivity would be suppressed. This is opposed to our observation.

The presence of oxygen vacancies is also unlikely because all the samples were slowly cooled down to room temperature with the oxygen partial pressure around 350 Torr. To further relieve the referee's concern about the oxidizing activity of Ca metal, we also performed XAS measurements and additional control experiments, please refer to the next response.

2. Related with the above comment, oxygen spectra in XPS and XAS should be checked

carefully.

Response:

Thank you for this suggestion. We would like to point out that the extremely surface-sensitive XPS cannot be used to probe possible oxygen vacancies in LSMO block, because CIO block was grown on top of the LSMO block in the heterostructures. Although XAS is also surface-sensitive, the probe depth is of a few nanometers. We thus conducted XAS spectra at the oxygen *K*-edge to investigate the potential presence of oxygen vacancies in LSMO.

Figure R5| The XAS at the oxygen *K*-edge of *n*CIO/20LSMO heterostructures and LSMO-LSAT film.

The XAS data at the oxygen *K*-edge of *n*CIO/20LSMO heterostructures and LSMO-LSAT film are presented in Fig. R5. The pre-edge spectral region spanning from 529 to 533 eV signifies the Mn *3d* unoccupied states through O_{2p}-Mn_{3d} orbital

hybridization, while the second band of Mn 3d at around 533 eV (indicated by the dashed line) is sensitive to the Mn valence, reflecting a band of minority e_g character [PRB, 46,4511 (1992)]. In the presence of oxygen vacancies in the LSMO sample, a decrease in the total peak intensity (including both peaks at 529.4 and 532.3 eV) from 529-533 eV would be anticipated due to diminished O_{2p} - Mn_{3d} orbital hybridization. This phenomenon is usually observed in ABO_{3-x} perovskites with oxygen vacancies, such as $LaNiO_{3-x}$ [Adv. Mater. 30, 1705904 (2018)] and $LaCoO_{3-x}$ [Nat. Commun. 12, 1853 (2021)]. However, in our study, we did not observe a significant reduction in both peaks, leading us to conclude that the formation of oxygen vacancies during sample preparation is unlikely. The change in the relative intensity of the two peaks, on the other hand, unveils modifications of the O_{2p} - Mn_{3d} orbital hybridization due to different epitaxial strain and interfacial effects.

3. Density-functional theory calculation is missing in the manuscript, which is crucial to understand the mechanism.

Response:

Thank you for this suggestion. We note that there are a lot of DFT works on LSMO, and an on-site correlation U must be taken into consideration considering the strongly-correlated nature. The selection of U , however, sometimes is tricky. Additionally, there are clear signatures of phase separation in the heterostructures. It is extremely challenging to construct such a mesoscopic model especially when electronic

correlation has to be taken into consideration {Please refer to the Section 3.10 in [*Phys. Rep.* 344, 1 (2001)] for a detailed discussion of the technical challenge}. More importantly, the experimental data set is clear and is understandable using the well-established double-exchange theoretical model. We thus prefer not to involve a potentially tricky theoretical result in the current research.

4. In my opinion, other CaMeO_3 compounds can bring insulator-ti-metal transition in LSMO. Could you try this to see the results?

Response:

Thank you for this point. We would like to point out that the key of the insulator-to-metal transition is the CIO/LSMO interfacial effect on *B*-sites. In order to relieve the referee's concern, we have grown a $5\text{CaTiO}_3(\text{CTO})/20\text{LSMO}$ heterostructure, and the result is shown in Fig. R6. In stark contrast to the metallic state in $5\text{CIO}/20\text{LSMO}$, $5\text{CTO}/20\text{LSMO}$ is strongly insulating with the resistivity rapidly increases with decreasing temperature. From this control experiment, one then can conclude that oxidizing effect from Ca is negligible in the observed insulator-to-metal transition. We now include the control experiment in the supplementary materials.

Figure R6 | The ρ - T curves of 5CTO/20LSMO and 5CIO/20LSMO heterostructures.

We also would like to refer the XRD pattern of the two heterostructures in order to explain the lacking of a metallic state in the 5CTO/20LSMO heterostructure. As shown in Fig. R7, in contrast the interface-driven lattice expansion in 5CIO/20LSMO, the 5CTO/20LSMO has a smaller out-of-plane lattice parameter. This comparison also indicates that interface effect is weak in the 5CTO/20LSMO heterostructure, in consistent with its robust insulating state.

Figure R7 | The XRD pattern of 5CTO/20LSMO and 5CIO/20LSMO heterostructures.

5. How about the stability of the heterostructures in ambient oxygen pressure annealing?

Response:

Thank you for this comment. In order to test the stability of the sample, we prepared another 5CIO/20LSMO heterostructure, which was additionally annealed at 750°C, 350 Torr (O₂) for 30 min after growth. As shown in Fig. R8, the overall temperature dependence of resistivity of this control sample is about the same as the one without high-temperature annealing. This control experiment suggests that the heterostructures are stable against high-temperature annealing under a high partial pressure of oxygen.

Figure R8 | The ρ - T curve of 5CIO/20LSMO heterostructure.

Report of the third Referee

I co-reviewed this manuscript with one of the reviewers who provided the listed reports.

This is part of the Nature Communications initiative to facilitate training in peer review and to provide appropriate recognition for Early Career Researchers who co-review manuscripts.

Response:

We appreciate your efforts very much. We hope the responses and the revisions will relieve your concerns.

Report of the fourth Referee

The paper “Atomically controlled insulator-to-metal transition in strongly correlated iridate/manganite heterostructures” by Men et al. reports a study on $\text{La}_{0.67}\text{Sr}_{0.33}\text{MnO}_3$ thin films capped with different thicknesses (n in unit cells) of CaIrO_3 (CIO/LSMO). The resistivity, magnetization, XAS and synchrotron XRD measurements were done varying n from 0 to 5 unit cells. They report that the electronic transport evolves from insulating to metallic behavior when $n > 2$. The Curie temperature and magnetization also increase at around this thickness. The authors concluded that that the CIO thickness in CIO/LSMO heterostructure is critical to the observed phenomena, driven by a Mn^{4+} -to- Mn^{3+} charge transfer at the interface. XRD symmetric scans are reported in the SI for some the main samples of the studies. Reciprocal space mapping and STEM imaging is reported for different samples in Figure 1.

The authors claim novelty based on their observed metallicity in a heterostructure consisting only of correlated insulators. LSMO and CIO are both metals in their groundstate and can be made insulating by biaxial strain, e.g. on K_2TaO_3 . The authors do mention this but do not adequately rule out that their heterostructure is allowing one or more of the constituent materials to return to their bulk-like properties through a trivial effect such as strain relaxation. Other major concerns that could trivially lead to a similar result are CIO-capping effect on LSMO quality, strain-induced critical CIO thickness, and CIO/LSMO heterostructure quality based on STEM.

In its current form we do not find the manuscript acceptable for publication. It may be

acceptable after addressing our below concerns, approximately in order of importance:

Response:

Thank you for the efforts in reviewing our work. The criticisms are highly helpful in improving the quality of our manuscript. We next response the comments separately.

1. Sample characterization is lacking.

a. The STEM image in Figure 1c is too limited. Please provide a low magnification image and an image including KTO/LSMO interface (as is written in the text) and the sample surface.

Response:

Thank you for this comment. A low magnification image is provided in Fig. R1. Two interfaces are also shown to demonstrate the good quality of interfaces. The sample surface, on the other hand, is partially destroyed during TEM sample preparation. The sacrificed portion is about 3 u.c. estimated from the difference between the original thickness of the CIO block and the residual CIO thickness estimated from TEM image. One should be noted that sample damage is unavoidable in iridate samples.

Figure R1 | A large-area TEM image of the 30CIO/30LSMO heterostructure.

b. Why are there intensity modulations and some diagonal stripes in the CIO layer?

Response:

We would like to point out that it is challenging to perform TEM measurement on ultrathin iridate films because Ir elements are easily lost during TEM sample preparation. For instance, unusual regions, such as the diagonal stripes as the referee pointed out, are typical features related to sample damage during TEM sample preparation. A similar observation was also reported on other iridate samples, e.g., in [Adv. Mater. 33, 2008269 (2021); Adv. Mater. 34, 2109163 (2022)].

c. Provide a STEM image for one of the main samples in the study, n=2 or 3 for instance.

The one currently shown is n=30 which, according to Figure 4 is relaxed. We suspect

CIO does not grow so well on fully strained LSMO.

Response:

Following your suggestion, and considering the above-mentioned technical challenging (i.e., the top 3 u.c. iridate region is unavoidably destroyed) in preparing TEM samples on iridate thin films, we prepared a 4CIO/18LSMO heterostructure. The total thickness of this heterostructure is close to that of 2CIO/20LSMO as well as the 3CIO/20LSMO. As a result, epitaxial strain of these three heterostructures are similar. Moreover, in order to further protect the iridate block, the heterostructure was capped with a SrTiO₃(STO) block of 10 u.c.

Figure R9 | Several representative TEM images of the 10STO/4CIO/18LSMO heterostructure.

As shown in Fig. R9, we present TEM images of several regions, where both the LSMO/KTO interface and the CIO/LSMO interfaces are clear to be seen. The good

quality of interface is in consistent with the observation on the 30CIO/30LSMO heterostructure. We would like to point out that this consistency is reasonable. For the heterostructure with a thick CIO block, even the sample surface is destroyed during FIB preparation, the sufficiently thick iridate block still preserves the interface region. In other words, the interface quality should be independent on n considering the bottom-to-top growth method. We now included the TEM data of the 10STO/4CIO/18LSMO heterostructure in the supplementary materials.

d. Interfacial intermixing is very likely occurring. Especially for $n=1, 2$, and given the high energy of the PLD growth parameters, it is unlikely that the interface is sharp, or even distinguishable for these low thicknesses. STEM-EELS is essential in ruling out interfacial intermixing, please provide that in addition to the HAADF images.

Response:

We would like to refer to the last response for the similar interface quality in the CIO/LSMO heterostructures. We have performed a comprehensive EELS analysis on the interface of the 30CIO/30LSMO heterostructure. As shown in Fig. R2, the EELS spectra of Ir/Mn as well as Ca/La indicate that the chemical intermixing crossing the interfaces is around one unit cell. This observation allows us to conclude that interfacial intermixing plays a minor role in the main observation, i.e., an insulator-to-metal transition driven by interfacial effect.

Figure R2 | The updated EELS data of the 30CIO/30LSMO heterostructure. The grey zone indicates the interface region.

e. The reciprocal space map in Figure 1b is only of a LSMO film, not one of the bilayers studied in the rest of the paper. An RSM including CIO is essential in ruling out strain relaxation in the CIO cap. Here we accept a thicker CIO layer because probably $n < 5$ will not be visible.

Response:

Thank you for this point. We have performed RSM measurements on heterostructures with $n=0, 2, 3, 5,$ and 30 . As can be seen from Fig. R10, all the LSMO blocks are fully strained, unveiling a negligible strain relaxation of LSMO in the n CIO/20LSMO heterostructures. As expected, the CIO layer is unobservable for $n < 5$, and the CIO layer in the 30CIO/20LSMO is clearly relaxed.

Figure R10 | RSMs of n CtO/20LSMO ($n=0, 2, 3, 5, 30$ -unit cells) heterostructures.

In order to rule out strain relaxation effect in the CtO block, we have prepared 30 u.c. and 50 u.c. CtO single films on KTO. Note that the total thickness of the 30CtO is thicker than the metallic 3CtO/20LSMO heterostructure, while the thickness of the 50CtO film is comparable with that of the 30CtO/20LSMO heterostructure. In other words, strain relaxation in CtO layer would be more substantial in 30CtO film as compared to the 3CtO/20LSMO heterostructure, and strain relaxation effect should be similar in the 50CtO film and 30CtO/20LSMO heterostructure. Nevertheless, as shown in Fig. R11, while the 50CtO film is more conductive than the 30CtO film, both CtO films are strongly insulating, in stark contrast to the metallic state in 3CtO/20LSMO as well as 30CtO/20LSMO heterostructures. We thus rule out potential effect of strain relaxation in the CtO layer in the emergent metallicity of the heterostructures.

Figure R11 | The ρ - T curves of 30C1O and 50C1O films.

The updated RSMs are now included in the supplementary materials.

2. The electrostatic force microscopy images (Supplementary Figure 2) look like they are dominated by scanning artifacts. E.g. the horizontal streaks.

Response:

Following the referee's suggestion, we have repeated the EFM measurements and eliminated the scanning artifacts.

a. The repeated structures that are light on the right and dark on the left suggest that the signal is not real – in the reverse trace does the image look the same?

Response:

We agree with the referee that the streaks are artifacts. However, as shown in Fig. R12, the updated EFM images (without the scanning artifacts) support the same conclusion in the previous version. Explicitly speaking, a cluster-like feature is

observed on the 20LSMO film while the electric conductivity is much more uniform in the 3CIO/20LSMO heterostructure.

Figure R12| Electrostatic force microscopy images of LSMO film (left panel) and 3CIO/20LSMO (right panel) heterostructure.

b. We think the images mostly reflect a very rough surface topography. Is it possible to provide just AFM topography image and quantify roughness?

Response:

According to your suggestion, we have performed AFM measurements on all the n CIO/20LSMO heterostructures and the LSMO-LSAT film. Please refer to Fig. R4 for the updated AFM images. The roughness is summarized in Table R1, which demonstrates that all the samples have similar roughness, and the different EFM images in Fig. R11 are dominated by the difference in electric conductivity of the two samples.

Figure R4 | Atomic force microscopy images of n CIO/20LSMO heterostructures and LSMO-LSAT film.

Table R1 | The roughnesses of n CIO/20LSMO heterostructures and LSMO-LSAT film.

	$n = 0$	$n = 1$	$n = 2$	$n = 3$	$n = 5$	LSMO-LSAT
R_q (nm)	0.639	0.991	0.710	0.765	0.669	0.652
R_a (nm)	0.481	0.722	0.499	0.378	0.517	0.513

3. The authors should rule out a trivial capping effect.

a. This scenario seems possible based on the EFM images in SI Figure 2. Both images have similar bumps so they are probably originating in the LSMO. But the right image has 3 u.c. of CIO on top and the surface does look a little smoother. It therefore seems likely that the CIO capping layer is acting to preserve the LSMO quality. Possibly

suppressing strontium migration to the surface. The authors should comment.

Response:

This is a good point. We would like to refer the updated AFM images to clarify that the sample surfaces are similarly flat in all the heterostructures. The smoother EFM of 3CIO/20LSMO originates from the increased size of metallic clusters as well as connections between these clusters under the interfacial effect.

Next, it is reported that Sr-segregation is more serious in thicker films [*PRB* 77,085401 (2008)]. If the CIO plays a role in preventing Sr-segregation, more CIO is needed in thicker LSMO. Along this line, one should observe that the threshold thickness of CIO for triggering the insulator-to-metal transition increases with the thickness of the LSMO block. This is clearly opposite to our observation. The potential contribution from Sr-segregation thus can be safely excluded.

We add a comment on Page 19, “The reduced critical n with increasing LSMO thickness also rules out possible contribution from Sr-segregation, which would otherwise give rise to an opposite trend due to escalated Sr-segregation in thick LSMO films⁵³”.

b. Have they ever tried an “inert” capping layer that shouldn’t induce charge transfer? LaAlO₃ or SrTiO₃ for instance? If there really is physics driving the observed capping layer thickness-dependent behavior then this would be an easy way to show it.

Response:

This is a very helpful comment.

We have prepared a $5\text{CaTiO}_3(\text{CTO})/20\text{LSMO}$ heterostructure. As shown in Fig. R6, the $5\text{CTO}/20\text{LSMO}$ is strongly insulating in opposite to the metallic state in $5\text{CIO}/20\text{LSMO}$.

Figure R6 | The ρ - T curves of $5\text{CTO}/20\text{LSMO}$ and $5\text{CIO}/20\text{LSMO}$ heterostructures.

We also grown a $3\text{CIO}/5\text{STO}/20\text{LSMO}$ heterostructure to spatially separate iridate and manganite. As shown in Fig. R13, in contrast to the metallic behavior of the $3\text{CIO}/20\text{LSMO}$ heterostructure, the $3\text{CIO}/5\text{STO}/20\text{LSMO}$ heterostructure is insulating. This comparison strongly suggests that the CIO/LSMO interface plays a dominant role in driving the insulator-to-metal transition. We now add the control experiment in the supplementary material.

Figure R13 | The ρ - T curves of 3CfO/5STO/20LSMO and 3CfO/20LSMO heterostructures.

4. The authors say that the difference in MR between their capped LSMO and the “bulk” LSMO is what proves that they have an “emergent” metallicity and not just bulk-like metallicity. In particular they see a large MR at low T with 3 u.c. of CfO and a small MR when “bulk-like”. This is probably the most important data of the paper so they should show the CfO thickness-dependence. Looking at the resistivity, $n=2$ should be measurable at least. The MR behavior might be coming from CfO, this should rule it out.

Response:

Thank you for this point. Following your suggestion, we measured MR of n CfO/20LSMO at 50K and 200K. As shown in Fig. R14a, the low-temperature MR first increases then decreases with n , giving rise to the maximized value at $n = 2$. In Fig. R14b, where MR at 200K is presented, we observed a similar non-monotonic

dependence on n with $n = 2$ maximizes the MR strength. The intriguing n -dependent MR is summarized in Fig. R14c, from which one can see that the heterostructures feature a much larger MR than the bulk-like LSMO film. The interface-driven enhancement of MR is maximized around $n = 2$, which probably optimizes phase separation for promoting the MR effect.

Figure R14 | (a) MR of n CfO/20LSMO ($n = 2, 3, 5, 10$) heterostructures at 50 K. (b) MR of n CfO/20LSMO ($n = 0, 1, 2, 3, 5, 10$) heterostructures at 200 K. (c) CIO thickness dependence of the negative value of MR. The solid line is a guideline. The star symbol represents MR of LSMO-LSAT at 200 K.

In the updated manuscript, we add the important n -dependent MR in Fig. 1g. We also add a statement on Page 13 in order to elaborate this point, “Figure 2d also demonstrates that 2CfO/20LSMO is at the boundary of the insulator-to-metal transition. This state is special where the size of ferromagnetic clusters may be optimized, such that the clusters can be easily connected under magnetic fields while the field-induced resistivity change is still substantial considering the non-metallic state under zero field, leading to the maximized MR in Fig. 1g.”

In addition, the non-monotonic n -dependence of MR also indicates that the MR is not dominated by the CIO layer, which otherwise would lead to a linear dependence of MR on n . In order to further rule out the contribution from CIO layer, we measured MR of CIO films (30 u.c. and 50 u.c.) at various temperatures. As shown in Fig. R15, MR of the CIO films is negligible at high temperatures, and the low-temperature MR is about three orders of magnitude smaller than that of the heterostructures. This striking difference allows us to confidently exclude the MR contribution from CIO.

Figure R15 | MR of 50 u.c. and 30 u.c. CIO films at different temperatures. Note that the 30CIO film is rather insulating, such that a small temperature fluctuation during field scan leads to an artificial hysteresis.

In order to elaborate this point, we add a comment “The non-monotonic dependence of the MR on n (Fig. 1g), indicates that large MR of the heterostructure is not an inherent property of the CIO block either. Indeed, MR effect of a CIO single crystal diminishes quickly with increasing temperature¹⁵, and MR effect of CIO films

is almost unobservable at high temperatures²⁸.” on Page 7. The MR data of CIO films is now added in the supplementary materials.

5. The XRD data:

a. The symmetric XRD scans shown in SI Figure 1: Why is the CIO peak not visible? $n=10$ should certainly be thick enough. Are the authors not concerned that their CIO is not crystalline?

Response:

Thank you for this point. We would like to clarify that CIO peak is not visible because of two factors. Firstly, CIO is too thin. As shown in Fig. R16, the CIO film peak can only be barely seen when $n = 10$.

Figure R16 | θ - 2θ patterns of n CIO/20LSMO ($n=0, 1, 2, 3, 5, 10$ -unit cells) heterostructures.

Secondly, the lattice parameters between CIO (3.868Å) and LSMO (3.876 Å) are close to each other. In order to fully distinguish their XRD peaks, one has to introduce different epitaxial strain in CIO and LSMO. To this end, we have prepared a 30CIO/20LSMO heterostructure. RSM measurement shown in the supplementary Fig.8 demonstrates that the LSMO is fully strained while the CIO layer is relaxed. The large tensile strain reduces the lattice parameter of LSMO while the lattice parameter of CIO is bulk-like. In this context, the (002) peaks of LSMO and CIO are discernable, as shown in Fig. R17. Therefore, from the updated XRD data, we confirm that CIO block is crystalline in the heterostructures.

Figure R17| The XRD pattern of the 30CIO/20LSMO heterostructure.

b. Looks like the c lattice parameter is increasing with n. This could be due to a gradual relaxation of the tensile strain. As mentioned above, RSMS of the whole series are needed to rule this out. Otherwise it looks like the LSMO is returning to its bulklike properties.

Response:

The c -axis lattice parameter indeed increases with n (please refer to Page 16 in the manuscript). However, we would like to clarify that this phenomenon is due to the enhanced out-of-plane angle under the interfacial effect.

Following your suggestion, we have performed RSM measurement for all the n CIO/20LSMO heterostructures. Please refer to Fig. R10 (also Supplementary Fig. 8) for the updated RSM images, where one can see that the LSMO block is fully strained in all the heterostructures.

c. Also, the LSMO peak intensity diminishes with increasing CIO. Could this also explain the reduction in intensity, with increasing cap thickness, of the half-order peaks shown in Figure 3b-c?

Response:

Thank you for the crucial comment. We would like to clarify the $00L$ and half-order peaks probe different ions in solids. Specifically, the former and the latter are dominated by the arrangement of cations and anions (oxygen in oxides), respectively [*Acta Cryst. A31*, 756 (1975); *Acta Cryst. B28*, 3384 (1972)].

The heterostructures with an ultrathin CIO layer disrupt the arrangement of cations while preserving the ordering of anions. Because of the different atomic factors of cations in CIO and LSMO, the diffraction intensity of the $00L$ peak is reduced in the heterostructures. Similar results can also be observed on nickelate films with a capping, e.g., in *arXiv:2403.00960* and *J. Phys. D: Appl. Phys. 56 024003 (2023)*. As a comparison, both CIO and LSMO have the same anion, i.e., oxygen. In other words,

the structural factors of anions are the same in CIO and LSMO. The intensity variation of the half-order peaks thus can only be ascribed to the modulated magnitude of octahedral rotation/tilting.

The intensity reduction of the rotation peaks (Fig. 3b), which points to an enlarged out-of-plane bond angle, is also consistent with the enlarged *c*-axis lattice parameter, as the referee pointed out in the last comment.

d. In Figure 3b-c it looks like there are fits to the data. Are these fits, and if so, what are the fit parameters? The FWHM would be especially important to show as evidence that the structural distortion is constant over the whole film, and constant with increasing *n*.

Response:

Thank you for this point. The solid lines are guidelines of the data. Nevertheless, we tried to fit the data with the Gauss function in Fig. R18. The extracted FWHM of the heterostructures are 0.085 ± 0.015 r.l.u.. We would like to clarify that the FWHM of the half-order peaks characterizes the coherent length of the distortion. The similar FWHM indicates that the coherent length of distortion is about the same in all the heterostructures. This is reasonable because all the heterostructures were prepared under the same condition and the crystalline quality is comparable.

Figure R18 Room-temperature synchrotron XRD around the (0.5 0.5 1.5) and (0.5 1.5 1.5) Bragg reflections, respectively. The solid line is fitted data by the Gauss function.

We would like to clarify that the magnitude of octahedral rotation/tilting is proportional to the square root of the half-order peaks [*Acta Cryst. A*31, 756 (1975); *Acta Cryst. B*28, 3384 (1972)]. Therefore, the intensity modulation in Fig. R18 confirms modulation of octahedral rotation/tilting in the heterostructures. In order to elaborate this point, we add a statement “Note that the peak intensity is proportional to the squared of octahedral tilting^{50,51}, this observation thus explicitly points out that the out-of-plane Mn-O-Mn bond angle has been systematically enhanced in the CIO/LSMO heterostructures as compared to the 20LSMO film.” on Page 15.

6. The findings of Figure 4, that a partially relaxed LSMO needs less CIO cap in order to become metallic, seems to support our suspicion that strain (relaxation) of one or both of the layers play a larger role than interfacial charge transfer. Have the authors not tried thinner LSMO?

Response:

We would like to clarify that slightly increasing the LSMO thickness can not give rise to a similar insulator-to-metal transition in heterostructures. For example, the total thickness of the 3CIO/20LSMO is smaller than the 30LSMO film but the former is metallic while the latter is insulating.

Please also refer the RSMs of the n CIO/20LSMO, which unveiled that LSMO layer is fully strained in all the heterostructures. This observation excludes the role of strain relaxation of LSMO in triggering the insulator-to-metal transition.

Figure R11 | The ρ - T curves of 30CIO and 50CIO films.

Similarly, a possible strain relaxation of CIO also fails to explain the insulator-to-metal transition. Explicitly, we have prepared a CIO film of 30 u.c. to mimic the potential strain relaxation in 10CIO/20LSMO considering the similar total thickness. As shown in Fig. R11, the CIO film is insulating and has a much higher resistivity than the heterostructures. This demonstrates that strain relaxation of CIO can not account

for the insulator-to-metal transition, either.

Figure R19 | The ρ - T curve of a 30CIO/10LSMO heterostructure.

Following your suggestion, we have grown a series of n CIO/10LSMO heterostructures. As shown in Fig. R19, it can be seen that the heterostructure is quite insulating even when the CIO capping is 30 unit cells and the CIO block is partially relaxed. This observation is understandable from the schematic illustration shown in Fig. R3 of the interface-driven percolation. When LSMO is too thin, the metallic clusters are too small to be connected by the interfacial effect, leading to a robust insulating state in the n CIO /10LSMO heterostructures.

Figure R3| The schematical diagram of the percolation-driven insulator-to-metal transition. The conductive path is highlighted by red curves.

7. If we extrapolate Figure 4 to lower a lattice parameter then we would conclude that this “emergent” metallic state of LSMO should be stabilized with no capping layer, is this the case?

Response:

We would like to emphasize that a sufficiently small lattice parameter indicates that the LSMO is fully relaxed. Such a relaxed LSMO will return to the bulk-like metallic state (like LSMO-LSAT film) due to a trivial strain relaxation effect. However, the metallic state of bulk-like LSMO film is different from the metallic state in the heterostructure in many aspects as discussed in the manuscript. For instance, the bulk-like LSMO-LSAT displays no MR hysteresis while the metallic heterostructures have a large MR hysteresis as shown in Fig. 1.

8. The authors make volume fraction arguments based on the low temperature magnetization, noting that the 1 u.c. CIO cap increases M by 20% while the volume fraction of CIO is only 5%. As the authors have not provided evidence that there is no cation intermixing and the high energy growth probably leads to 2-3 unit cells of intermixing, the volume fraction of the interface could be $3/18 = 17\%$. This matches quite well with the magnetic volume fraction determination. The authors need to address this possibility.

Response:

Thank you for this point. We would like to refer to the updated EELS data in Fig. R2, where we demonstrate that chemical intermixing is around \sim one unit cell.

In addition, we also tried the way how the referee estimates the volume fraction of interface for other heterostructures. Taking the 2CIO/20LSMO for example, however, the estimated volume fraction of interface $4/18=22\%$ deviates significantly from the volume fraction $\sim 45\%$ estimated from magnetization. We wish the updated EELS data and the above analysis would relieve your concern about cation intermixing.

9. Would significant hysteresis be expected in the resistivity versus temperature if there is a percolation mechanism at play?

Response:

Thank you for this comment. Following your suggestion, we performed additional resistivity measurements on two representative heterostructures, 1CIO/20LSMO

(insulating) and 5CIO/20LSMO (metallic). As shown in Fig. R20, we found there is no observable temperature-dependent hysteresis on both samples.

Figure R20 | The cooling and warming ρ - T curves of 1CIO/20LSMO (left panel) and 5CIO/20LSMO (right panel) heterostructures.

This observation demonstrates that the heterostructures are distinct from conventional phase-separated manganites. Explicitly speaking, the percolation is driven by an interfacial effect in the heterostructures, while a temperature-driven percolation effect is usually observed in manganites. The latter usually associates with a first-order transition and varying temperature increases or decreases the size of clusters. Therefore, a temperature-dependent hysteresis is typically observed when one scanning temperature back and forth. Similarly, if one can manipulate the interface effect back and forth, an *interface-dependent hysteresis* would be observed. However, the interface effect is sample-dependent and there is no parameter can be manipulated in order to control this effect back and forth in a single sample.

10. More information is needed on how charge transfer was estimated from the XAS.

Details on peak fitting, parameters, goodness of fit and uncertainty on the charge transfer value itself should be provided.

Response:

To estimate the charge transfer value at the heterostructures, we employed multiplet calculations for Mn^{3+} and Mn^{4+} to conduct quantitative fittings and determine the mean valence of Mn in CIO/LSMO heterostructures, as illustrated in Fig. R21. By utilizing the configuration interaction cluster model and parameters from a previous study [*Nat. Commun.*, 12, 3136 (2021)], we performed calculations for Mn^{3+} and Mn^{4+} using CTM4XAS [*Coord. Chem. Rev.* 249, 31 (2005), *Micron* 41, 687 (2010)]. The calculated spectra closely matches the experimental results for Mn^{3+} from LaMnO_3 and Mn^{4+} from Li_2MnO_3 .

Figure R21 | The fitting of XAS at Mn *L*-edge with calculated Mn^{4+} , Mn^{3+} spectra using the configuration interaction cluster model .

By combining these two spectra, we were able to quantitatively reproduce our experimental spectra of CIO/LSMO heterostructures with $n=0$ and $n=5$ as 50% Mn^{4+} + 50% Mn^{3+} and 40% Mn^{4+} + 60% Mn^{3+} , respectively. The XAS profiles of samples with $n=1, 2,$ and 3 are nearly identical to that of the $n=5$ sample. Consequently, we estimated the mean valence of 3.5 and 3.4 for Mn in the $n=0$ and $n=1, 2, 3, 5$ samples, respectively. This allowed us to determine a charge transfer value of 0.1 e/Mn . The determined valence of $\text{Mn}^{3.5+}$ for the LSMO film slightly exceeded the expected 3.33+ from the chemical formula $\text{La}_{0.67}\text{Sr}_{0.33}\text{MnO}_3$. This discrepancy may stem from the covalent Mn-O interactions in the calculated Mn^{4+} and Mn^{3+} spectra. The relative intensity between 640.3 and 642.1 eV, which are the crucial features for Mn^{4+} and Mn^{3+} , was used to assess the fitting quality. The fitting uncertainty is approximately $\pm 0.03 e/\text{Mn}$. We now included the fitting details in the supplementary materials.

11. Do the authors have any comment on the kinks at $\sim 120\text{K}$ and $\sim 200\text{K}$ in Fig. 1d and Fig. 2a metallic curves?

Response:

This is an interesting point. In manganites, a resistivity kink may be observed around the magnetic transition temperature. However, the kink temperature in our heterostructure is much lower than the magnetic transition temperature. Currently, we don't have a clear idea on the physics behind the kink.

12. What thickness do the authors assume when calculating the resistivity, that all the

LSMO becomes metallic and nothing else?

Response:

Thank you for this comment. In the original manuscript, the resistivity was calculated only with the LSMO thickness considering the fact that CIO layer is much more insulating than the LSMO layer. We also calculated the resistivity with the total thickness of the heterostructures. As shown in Fig. R22, one can see that the n -dependence of the resistivity is the same as Fig. 2a in the manuscript.

Figure R22 | The ρ - T curves of n CIO/20LSMO heterostructures calculated with the total thickness.

We added “The resistivity of each heterostructure was calculated by only taking into consideration of the thickness of the LSMO block.” on Page 11 in the caption of Fig. 2.

13. SI Figure S4 makes the saturation regime clear but not the coercive field, please show a close-up of the low field data.

Response:

Following your suggestion, we present the enlarged M - H curves in the updated SI, see Fig. R23 for example. The coercivity field of the heterostructures is about one order of magnitude larger than that in the LSMO-LSAT film. The enhanced coercivity field is in line with the phase separation scenario of the heterostructures. Explicitly speaking, in materials composed of well-separated ferromagnetic clusters, the coercivity is enhanced because of the enhanced magnetic domain rotation. In this sense, it is reasonable that the CIO/LSMO heterostructures consisting of tiny ferromagnetic clusters show larger coercivity fields.

Figure R23 | Enlarged M - H loops of n CIO/20LSMO heterostructures and LSMO-LSAT film.

14. Can the authors add arrows to Figure 2b, similar to 2c, showing the inflection points that they discuss in the text?

Response:

According to your suggestion, we used hollow circles to highlight the inflection points in Fig. 2b. And we added “The hollow circles indicate the inflection points of the heterostructures.” on Page 11 in the caption.

15. In Fig. 2d, could they add mention if the line is a guide to eye or something else?

Response:

Thank you for pointing that out. The line is a guideline, and we added “The black line is a guideline.” on Page 11 in the caption.

16. Please add the field direction to Figure 1e-f.

Response:

The field direction is in-plane and this detail is updated on Page 5 in the caption.

17. Remove the word “alternating” from the 2nd paragraph of “results and discussion” – it implies a repeating superlattice structure but these samples are just bilayers.

Response:

According to your suggestion, we removed the word in the updated manuscript.

Report of the fifth Referee

I co-reviewed this manuscript with one of the reviewers who provided the listed reports.

This is part of the Nature Communications initiative to facilitate training in peer review and to provide appropriate recognition for Early Career Researchers who co-review manuscripts.

Response:

Thank you very much for the careful review on our work. We have responded relevant comments one-by-one, we wish the control experiments as well as the updated data will relieve your concerns.

Next, we list all other major changes which were not mentioned in the above responses.

1) Page 2, 1st paragraph.

Replace “*Such a drastic transition is due to the cooperative interplay of electronic variation and structural modification thanks to a substantial charge transfer at the Ir/Mn interface.*” with “Such a drastic transition is induced by an interfacial charge transfer, which interestingly alters the electronic and crystalline structures of the bulk region rather than the limited ultrathin interface in a cooperative manner.”.

2) Full text.

Replace “*electronic correlation*” with “effective correlation”.

3) Page 3, 1st paragraph.

Add “sample design”.

4) Page 5, Fig. 1 caption.

Add “Electron energy loss spectroscopy (EELS) spectra of B-site ions (Ir and Mn) and A-site ions (Ca and La) across the CIO/LSMO interface. The interface region is marked by a grey zone.”.

5) Page 5, Fig. 1 caption.

Add “CIO thickness dependence of MR at 5 K and 200 K. MR of the LSMO-LSAT film at 200 K is also shown for comparison.”.

6) Page 6, 1st paragraph.

Replace “*atomically sharp with negligible chemical intermixing.*” with “well-identified²⁸”.

7) Page 6, 2nd paragraph.

Replace “*Fig. 1d*” with “Fig. 1e”.

8) Page 7, 1st paragraph.

Add “a”.

9) Page 7, 1st paragraph.

Replace “*Fig. 1d*” with “Fig. 1e”.

10) Page 7, 1st paragraph.

Delete “*The large hysteresis was not observed on the 20LSMO film (Fig. 1f).*”.

11) Page 7, 1st paragraph.

Replace “*property*” with “properties between 3CIO/20LSMO and the 20LSMO film”.

12) Page 12, 1st paragraph.

Replace “*Considering the fact that phase separations are frequently observed in insulating manganites^{35,36}, it is highly potential that phase separation also exists in the heterostructures such that*” with “Indeed, a broad transition is typically observed in phase-separated manganites^{16,44}”.

13) Page 12, 1st paragraph.

Delete “*Moreover, a large MR hysteresis like the one in Fig. 1e is also believed to be the hallmark of phase separation in manganites^{35,37,38}*”.

14) Page 14, 1st paragraph.

Replace “*641.5 eV*” with “640.3 eV”.

15) Page 17, 2nd paragraph.

Replace “*The*” with “In each series, the”.

16) Page 17, 2nd paragraph.

Replace “*indicating a suppressed electronic correlation as also unveiled from the reduced in-plane lattice parameter in Fig. 4b.*” with “demonstrating a destabilized insulating state. The decreased in-plane lattice parameter in Fig. 4b confirms that the effective correlation is reduced thanks to the partially relaxed tensile strain.”.

17) Page 19, 1st paragraph.

Add “With the further reduced effective correlation due to enhanced strain relaxation”.

18) Page 19, 2nd paragraph.

Add “The efficient percolation transition was unveiled to be driven by an intriguing Ir/Mn interfacial effect, which alters both electronic and crystalline structures of the manganite block in the bulk region rather than the ultrathin interface region.”.

19) References.

Add references:

[18] Qi, J. et al. Strain Modified Oxygen Evolution Reaction Performance in Epitaxial, Freestanding, and Van Der Waals Manganite Thin Films. *Nano Lett* 22, 7066-7072 (2022).

[21] Yi, D. et al. Tuning Perpendicular Magnetic Anisotropy by Oxygen Octahedral Rotations in $(\text{La}_{1-x}\text{Sr}_x\text{MnO}_3)/(\text{SrIrO}_3)$ Superlattices. *Phys. Rev. Lett.* 119, 077201 (2017).

[22] Okamoto, S. et al. Charge Transfer in Iridate-Manganite Superlattices. *Nano*

Lett. 17, 2126-2130 (2017).

[23] Huang, X. et al. Novel Spin–Orbit Torque Generation at Room Temperature in an All-Oxide Epitaxial $\text{La}_{0.7}\text{Sr}_{0.3}\text{MnO}_3/\text{SrIrO}_3$ System. *Adv. Mater.* 33, 2008269 (2021).

[24] Rondinelli, J. M. & Spaldin, N. A. Structure and Properties of Functional Oxide Thin Films: Insights From Electronic-Structure Calculations. *Adv. Mater.* 23, 3363-3381 (2011).

[29] Biswas, A. & Jeong, Y. H. Persistent semi-metal-like nature of epitaxial perovskite CaIrO_3 thin films. *J. Appl. Phys.* 117, 195305 (2015).

[30] Hirai, D., Matsuno, J., Nishio-Hamane, D. & Takagi, H. Semimetallic transport properties of epitaxially stabilized perovskite CaIrO_3 films. *Appl. Phys. Lett.* 107, 012104 (2015).

[48] Ghising, P. & Hossain, Z. Electric field control of the photoinduced resistance increase in $\text{La}_{0.7}\text{Sr}_{0.3}\text{MnO}_3/\text{LaTiO}_3/\text{SrTiO}_3$ heterostructure. *Phys. Rev. B* 100, 115119 (2019).

[49] Kaneta-Takada, S. et al. Giant spin-to-charge conversion at an all-epitaxial single-crystal-oxide Rashba interface with a strongly correlated metal interlayer. *Nat. Commun.* 13, 5631 (2022).

[50] Glazer, A. M. The classification of tilted octahedra in perovskites. *Acta Cryst.* B28, 3384-3392 (1972).

[51] Glazer, A. M. Simple ways of determining perovskite structures. *Acta Cryst.* A31, 756-762 (1975).

- [52] Loktev, V. M. & Pogorelov, Y. G. Peculiar physical properties and the colossal magnetoresistance of manganites (Review). *Low Temp. Phys.* 26, 171-193 (2000).
- [53] Herger, R. et al. Structure determination of monolayer-by-monolayer grown $\text{La}_{1-x}\text{Sr}_x\text{MnO}_3$ thin films and the onset of magnetoresistance. *Phys. Rev. B* 77, 085401 (2008).
- [54] Wang, W. et al. Electrical manipulation of skyrmions in a chiral magnet. *Nat. Commun.* 13, 1593 (2022).
- [55] Tang, J. et al. Magnetic skyrmion bundles and their current-driven dynamics. *Nat. Nanotechnol.* 16, 1086-1091 (2021).

Reviewers' Comments:

Reviewer #2:

Remarks to the Author:

The revision of the manuscript is acceptable for me. I recommend the publication of the manuscript at the current format.

Reviewer #4:

Remarks to the Author:

The authors have done an admirable job of responding to our comments and questions, including apparently growing new samples and performing new measurements. We like the control samples with different cap and buffer layers, and the thick ClO, we think these are important results to rule out the trivial effects we were concerned about. We also appreciate the new STEM-EELS data that shows that chemical intermixing is not extending beyond 1 unit cell at the interfaces. We also like the new presentation of the MR data and we think this makes their message more clear. We noticed the oxygen XAS data in the rebuttal file and were wondering why this is not included in the manuscript or the supplementary information? Does it not lend support to their argument of charge transfer since the pre-peak intensity diminishes when there is a ClO capping layer, indicating perhaps greater occupation of the Mn 3d and thus a reduced 3d-2p hybridization? Something to consider. Otherwise, the manuscript is acceptable in its current form.

Reviewer #5:

Remarks to the Author:

Report of the Second Referee

The revision of the manuscript is acceptable for me. I recommend the publication of the manuscript at the current format.

Response:

Thank you for your recognition of our work. Your comments and suggestions are of great help to our work.

Report of the third Referee

Response:

We appreciate your participation in reviewing this work. And thank you again for your efforts in the review process.

Report of the fourth Referee

The authors have done an admirable job of responding to our comments and questions, including apparently growing new samples and performing new measurements. We like the control samples with different cap and buffer layers, and the thick CIO, we think

these are important results to rule out the trivial effects we were concerned about. We also appreciate the new STEM-EELS data that shows that chemical intermixing is not extending beyond 1 unit cell at the interfaces. We also like the new presentation of the MR data and we think this makes their message more clear. We noticed the oxygen XAS data in the rebuttal file and were wondering why this is not included in the manuscript or the supplementary information? Does it not lend support to their argument of charge transfer since the pre-peak intensity diminishes when there is a ClO capping layer, indicating perhaps greater occupation of the Mn 3d and thus a reduced 3d-2p hybridization? Something to consider. Otherwise, the manuscript is acceptable in its current form.

Response:

Thank you for your recognition of the updated data and control experiments, and we are also highly appreciated for your positive recommendation. Please note that the manganite component is capped with the iridate component, the soft X-ray based oxygen XAS contains information in both the iridate and manganite components. While the oxygen XAS is sufficient to exclude the role of oxygen vacancy in the manganite component, it is tricky to draw a quantitative conclusion on the 3d-2p hybridization. We thus prefer not to include it in the supplementary materials.

Report of the fifth Referee

I co-reviewed this manuscript with one of the reviewers who provided the listed reports. This is part of the Nature Communications initiative to facilitate training in peer review and to provide appropriate recognition for Early Career Researchers who co-review

manuscripts.

Response:

We sincerely appreciate your efforts in reviewing our work.